# Learning Lightweight Object Detectors via Progressive Knowledge Distillation

## Abstract

Resource-constrained perception systems such as edge computing and vision-for-robotics require vision models to be both accurate and lightweight in computation and memory usage. Knowledge distillation is one effective strategy to improve the performance of lightweight classification models, but it is less well-explored for structured outputs such as object detection and instance segmentation, where the variable number of outputs and complex internal network modules complicate the distillation. In this paper, we propose a simple yet surprisingly effective sequential approach to knowledge distillation that progressively transfers the knowledge of a set of teachers to a given lightweight student. Our approach is inspired by curriculum learning: To distill knowledge from a highly accurate but complex teacher model, we construct a sequence of teachers to help the student gradually adapt. Our progressive distillation strategy can be easily combined with existing distillation mechanisms to consistently maximize student performance in various settings. To the best of our knowledge, we are the first to successfully distill knowledge from Transformer-based teacher detectors to convolution-based students, and unprecedentedly boost the performance of ResNet-50 based RetinaNet from 36.5% to **42.0%** AP and Mask R-CNN from 38.2% to **42.5%** AP on the MS COCO benchmark.

## 1 Introduction

The success of recent deep neural network models generally depends on an elaborate design of architectures with tens or hundreds of millions of model parameters. However, their huge computational complexity and massive memory/storage requirements make them challenging to be deployed in safety-critical real-time applications, especially on devices with limited resources, such as self-driving cars or virtual/augmented reality models. Such concerns have spawned a wide body of literature on compression and acceleration techniques. Many approaches focus on reducing computation demands by sparsifying/pruning networks (Lebedev & Lempitsky, 2016; Han et al., 2016), quantization (Rastegari et al., 2016; Wu et al., 2016), or neural architecture search (Zoph & Le, 2017; Liu et al., 2019), but reduced computation does not always translate into lower latency because of subtle issues with memory access and caching on GPUs (Tan et al., 2019; Ding et al., 2021).

Rather than searching over new architectures, we seek to better train *existing* lightweight architectures that have already been carefully engineered for efficient memory access. Instead of relying on additional data or human supervision, we follow the large body of work on knowledge distillation (Buciluǎ et al., 2006; Hinton et al., 2014) for compressing the information from a large model into a small model. While most recent efforts in knowledge distillation focus on image classification, relatively less work exists for distilling object detectors. The extension from classification to object detection and instance segmentation is nontrivial due to the complicated outputs of the tasks. Most detectors operate with multi-task heads (for classification, and box/mask regression) that can generate variable-length outputs. In the literature of detector distillation, recent work (Zhang & Ma, 2021; Shu et al., 2021; Yang et al., 2022b) mainly focuses on designing advanced distillation loss functions for transferring features from teachers to students. However, there are two unsolved challenges: 1) The *capacity gap* (Cho & Hariharan, 2019; Mirzadeh et al., 2020) between models can result in a sub-optimal distilled student even if the strongest teacher has been employed, which is undesired when optimizing the accuracy-efficiency trade-off of the student. Moreover, when trying to distill knowledge from Transformer-based teachers (Dosovitskiy et al., 2020; Liu et al., 2021)

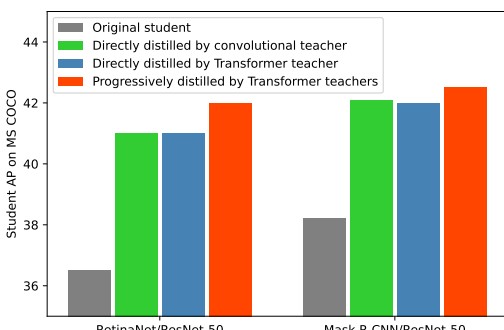

**Figure 1: Our proposed progressive distillation leads to state-of-the-art student performance.** When switching the teacher model from a convolution-based detector to a Transformer-based one with stronger detection performance, the student does not become more accurate, due to the architectural difference between the teacher-student pair. Progressively distilling knowledge from multiple teachers can mitigate the capacity gap and result in the best student performance.

to classical convolution-based students, this *architectural difference* can enlarge the teacher-student gap. 2) Current methods assume that one target teacher has been selected. However, this meta-level optimization of *teacher selection* is neglected in the existing literature of detector distillation. In fact, finding a pool of strong teacher candidates is easy, but trial-and-error may be necessary before determining one most compatible teacher for a specific student.

To address these challenges, we propose a framework to learn lightweight detectors through *progressive knowledge distillation*: 1) We find sequential distillation from multiple teachers arranged into a curriculum significantly improves knowledge distillation and bridges the teacher-student capacity gap. As shown in Figure 1, even with huge architectural difference, our progressive distillation can effectively transfer knowledge from Transformer-based teachers to convolution-based students, while previous methods cannot. 2) For the teacher selection problem, we design a heuristic algorithm for a given student and a pool of teacher candidates, to automatically determine the order of teachers to use. This algorithm is based on the analysis of the representation similarity between models, which does not require knowledge of the specific distillation mechanism to be used.

Overall, our progressive distillation is a *general meta-level* strategy that consistently improves both simple feature-matching distillation and more sophisticated ones. With the help of modern distillation mechanisms and teacher detectors, our progressive distillation learns lightweight RetinaNet and Mask R-CNN students with state-of-the-art accuracy. Furthermore, by analyzing the training loss dynamics of the student model, we find the improvement is *not* due to minimizing the training loss better. Rather, the knowledge transferred from multiple teachers can lead the student to a flat minimum, and thus help the student *generalize* better. To summarize, we transfer knowledge from multiple teachers to progressively distill a student. Our **main contributions** include:

- We propose a framework for learning lightweight detectors through progressive knowledge distillation, which is simple, general, yet effective. We develop a principled way to automatically design a sequence of teachers appropriate for a given student and progressively distill it.
- Our progressive distillation is a meta-level strategy that can be easily combined with previous efforts in detection distillation. We perform comprehensive empirical evaluation on the challenging MS COCO dataset and observe consistent gains.
- For the first time, we investigate distillation from Transformer-based teacher detectors to a convolution-based student, and find progressive distillation is the key to bridge their gap.
- We show the performance gain comes from better generalization rather than better optimization.

## 2 RELATED WORK

**Knowledge Distillation:** Knowledge distillation or transfer, an idea of training a shallow student network with supervision from a deep teacher, was originally proposed by Buciluǎ et al. (2006), and later formally popularized by Hinton et al. (2014). Different knowledge can be used, such as response-based knowledge (Hinton et al., 2014), and feature-based knowledge (Romero et al., 2015; Heo et al., 2019). Several multi-teacher knowledge distillation methods have been proposed (Vongkulbhisal et al., 2019; Sau & Balasubramanian, 2016), which usually use the average of logits and feature representations as the knowledge (You et al., 2017; Fukuda et al., 2017). Mirzadeh et al. (2020) find that an intermediate teacher assistant, decided by architectural similarities, can bridge the gap between the student and the teacher. We find it more effective to use a sequence of teachers instead of their ensemble, and extend Mirzadeh et al. (2020) to a more general case where teacher models have diverse architectures and their relative ordering is unknown.

**Object Detection and Instance Segmentation:** A variety of convolutional neural network (CNN) based object detection frameworks have been proposed, and can be generally divided into single-stage and two-stage detectors. Typical single-stage methods include YOLO (Redmon et al., 2016; Redmon & Farhadi, 2018) and RetinaNet (Lin et al., 2017b), and two-stage methods include Faster R-CNN (Ren et al., 2014), and Mask R-CNN (He et al., 2017). Recently, several multi-stage models are proposed, such as HTC (Chen et al., 2019a) and DetectoRS (Qiao et al., 2021). These detection frameworks achieve better detection accuracy with better backbone networks as feature extractors and with more complicated heads, which are more computationally expensive.

**Knowledge Distillation for detection and segmentation:** To reduce the computational cost, knowledge distillation has been used to develop efficient detectors. Chen et al. (2017) first use knowledge distillation to enforce the student detector to mimic the teacher's predictions. More recent efforts usually focus on learning from the teacher's features, rather than final predictions. Various distillation mechanisms have been proposed to leverage the impact of foreground and background objects (Wang et al., 2019; Guo et al., 2021), relation between individual objects (Zhang & Ma, 2021; Dai et al., 2021), or relation between local and global information (Yang et al., 2022a;b). Different from these methods that distill from a single teacher, we study distillation from multiple teachers, where a proper sequence of teachers is required, and we find a very simple feature-matching loss is adequate to significantly boost student performance.

## 3 APPROACH

We propose to progressively distill a student model $S$ with a pool of $N$ teachers $\mathcal{P} = \{T_i\}_{i=1}^N$. Typical object detectors are composed of four modules: (1) backbone, which extracts visual features, such as ResNet (He et al., 2016) and ResNeXt (Xie et al., 2017); (2) neck, which extracts multi-level feature maps from various stages of the backbone, such as FPN (Lin et al., 2017a) and Bi-FPN (Tan et al., 2020); (3) optional region proposal network (RPN), which is used in two-stage detectors; and (4) head, which generates final predictions for object detection and segmentation. We denote the output feature maps of the *neck* as $F^{\text{Net}}$, where Net can be either the student model $S$ or one of the teachers $T_i \in \mathcal{P}$. With neck modules like FPN, the feature maps can be multi-level.

We propose a meta-strategy for detector distillation that progressively learns a student using a sequence of teachers. To examine this meta-strategy without involving sophisticated distillation mechanisms, we introduce a simple feature-matching distillation for a single teacher $T_i$ in Section 3.1. Then we discuss progressive distillation with multiple teachers from $\mathcal{P}$ in Section 3.2.

### 3.1 SINGLE TEACHER DISTILLATION VIA SIMPLE FEATURE MATCHING

In order to learn a efficient student detector $S$ through distillation, we encourage the feature representation of a student to be similar to that of the teacher (Chen et al., 2017; Yang et al., 2021). To this end, we minimize the discrepancy between the feature representations of the teacher and the student. Without bells and whistles, we simply minimize the L2 distance between $F^{T_i}$ and $F^S$:

$$L_{\text{distill}} = \left\| F^{T_i} - r(F^S) \right\|_2^2, \tag{1}$$

where $r(\cdot)$ is a function used to match the feature map dimensions of the teacher and the student.

We define $r(\cdot)$ as follows:

- (Homogeneous case) If the numbers of channels and the spatial resolutions are both the same between $T_i$ and $S$, $r(\cdot)$ is an identity function.
- (Heterogeneous case) If the numbers of channels are different but the spatial resolutions are the same, we use $1 \times 1$ convolutional filters as $r(\cdot)$. If the spatial resolutions are different but the numbers of channels are the same, we use an upsampling layer as $r(\cdot)$.

Note that the mapping $r(\cdot)$ is only required at training time and thus *not adding any overhead* to the inference. Overall, our loss function can be written as:

$$L = \lambda L_{\text{distill}} + L_{\text{detect}}, \tag{2}$$

where $\lambda$ is a balancing hyper-parameter and $L_{\text{detect}}$ is the detection loss based on the ground truth labels. Compared to state-of-the-art detection distillation approaches (Zhang & Ma, 2021; Shu et al., 2021; Yang et al., 2022a;b), which introduce more complex designs of the distillation loss, this feature-matching distillation is simpler and does not require running the heads of the teacher model. Our distillation loss is illustrated in Figure 2-**Left**.

**Figure 2: Progressive knowledge distillation for object detectors. Left**: For each teacher-student pair, the training target is composed of two parts: $L_{\text{distill}}$ minimizes the discrepancy between the neck feature maps of the student and the current teacher, and $L_{\text{detect}}$ is the original detection loss based on the ground truth. **Right**: We use a *sequence* of teacher models to distill the lightweight student detector. The sequence of teachers forms a curriculum. Using a proper sequence of teachers can significantly boost the student model's performance. The example performance curve illustrates our method improves the COCO validation AP of ResNet-50 backboned RetinaNet student first from 36.5% to 37.9% using HTC (Teacher 1), and then from 37.9% to 39.9% using DetectoRS (Teacher 2).

## 3.2 Progressive Distillation with Multiple Teachers

The overall aim of knowledge distillation is to make a student mimic a teacher's output, so that the student is able to obtain similar performance to teacher's. However, the capacity of the student model is limited, making it hard for the student to learn from a highly complex teacher (Cho & Hariharan, 2019). To address this issue, multiple teacher networks are used to provide more supervision to a student (Sau & Balasubramanian, 2016; You et al., 2017). Unlike previous methods which distill knowledge from the ensemble of logits or features simultaneously, we propose to distill feature-based knowledge from multiple teachers *sequentially*. Our key insight is that instead of mimicking the ensemble of all feature information together, the student can be distilled more effectively by the knowledge provided by one proper teacher each time. This progressive knowledge distillation approach can be considered as designing a curriculum (Bengio et al., 2009) offered by a sequence of teachers, as illustrated in Figure 2-**Right**.

The crucial question is: *What is the proper order $\mathcal{O}$ of the teachers when distilling the student?* A brute-force approach might search over all orders and pick the best (that produces a distilled student with the highest validation accuracy). However, the space of permutation orders grows exponentially with the number of teachers, making this impractical to scale. Therefore, we propose a principled and efficient approach based on a correlation analysis of each model's learned feature representation.

First, we quantify the dissimilarity between each pair of models' representations, as a proxy for the capacity gap between them. Representation (dis)similarity (Raghu et al., 2017; Wang et al., 2018; Kornblith et al., 2019) has been studied to understand the learning capacity of neural models. In our setting, we find a linear regression model is adequate for measuring the representation dissimilarity. Given two pre-trained detectors A and B, we can freeze their parameters, and thus fixing the feature representations. Then we can learn a linear mapping $r(\cdot)$, implemented by a $1 \times 1$ convolutional layer at each feature level, as specified in the heterogeneous case in Section 3.1. $r(\cdot)$ is trained to minimize $L_{\text{distill}}$, so it can transform A's features to approximate B's features. After training $r(\cdot)$, we evaluate it by $L_{\text{distill}}$ on the validation set, and denote the validation loss as the *adaptation cost* $\mathcal{C}(A, B)$. This metric can be a proxy of the capacity gap between a pair of models: When $\mathcal{C}(A, B)$ is zero, a linear mapping can transform A's features to B's, and there is no additional knowledge from B. When $\mathcal{C}(A, B)$ is large, it is more difficult to adapt A's representation to B's. Note that the adaptation cost is non-symmetric – it is relatively easier to adapt a high-capacity model's representations to a low-capacity model's representations, than the other way around.

We design a heuristic algorithm to acquire a proper distillation order $\mathcal{O}$ automatically (details are shown in Algorithm 1 in the appendix). Suppose the maximum number of teachers to be selected is limited by $k$ (which can be arbitrarily decided according to desired training time), and we aim to find a teacher index sequence $\alpha$ no longer than $k$. We construct the teacher order backwards: The best performing teacher is set as the final target $T_{\alpha_k}$; before the final teacher, we use another teacher, which has the smallest adaptation cost $\mathcal{C}(\cdot, T_{\alpha_k})$ to that final teacher, as the penultimate teacher $T_{\alpha_{k-1}}$. We repeat this procedure to find preceding teachers, until: (1) when trying to select $T_{\alpha_j}$, we find the transfer costs from remaining teachers to the next teacher $\mathcal{C}(\cdot, T_{\alpha_{j+1}})$ are all larger than the transfer cost from the student to the next teacher $\mathcal{C}(S, T_{\alpha_{j+1}})$; or (2) we reach the given maximum step limit $k$. Intuitively, the resulting sequence of teachers bridges the gap between the student model and the teacher, with an increasingly difficult curriculum.

| Model | Input Res. | Backbone | Neck | Head | AP Box | AP Mask | Runtime (ms) |
|---|---|---|---|---|---|---|---|
| **Teachers** | | | | | | | |
| I | 1× | R50 | FPN | Mask R-CNN | 38.2 | 34.7 | 51 |
| II | 1× | R50 | FPN | FCOS | 38.7 | - | 36 |
| III | 1× | R50 | FPN | HTC | 42.3 | 37.4 | 181 |
| IV | 1× | R50+SAC | RFP | HTC (DetectoRS) | 49.1 | 42.6 | 223 |
| V | 1× | R50+SAC | RFP | Mask R-CNN | 45.1 | 40.1 | 142 |
| **Students** | | | | | | | |
| I | 1× | R50 | FPN | RetinaNet | 36.5 | - | 43 |
| II | 1× | R50 | FPN | Mask R-CNN | 38.2 | 34.7 | 51 |
| III | 1× | R18 | FPN | Mask R-CNN | 33.3 | 30.5 | 29 |
| IV | 0.25× | R50 | FPN | Mask R-CNN | 25.8 | 23.0 | 17 |

**Table 1: Configuration and COCO performance of the student and teacher detectors.** We investigate a variety of models with heterogeneous input resolutions, backbones, necks, and head structures. '1×' input resolution refers to the standard $1333 \times 800$ resolution, and '0.25×' means $333 \times 200$ resolution. 'R-' backbones are ResNets with different number of layers.

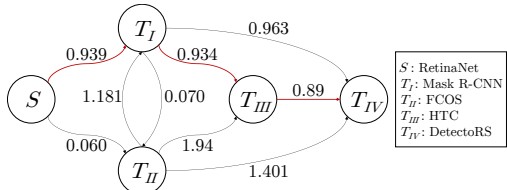

**Figure 3: Adaptation costs among models.** The number on each directed edge is the adaptation cost metric described in Section 3.2. Some edges are not shown for visual clarity. The red path is suggested by our proposed Algorithm 1 when $k = 3$ teachers are selected: (1) use the best performing Teacher IV as the final teacher in the sequence, (2) use the teacher closest to Teacher IV, which is Teacher III, as the second teacher, and (3) use the teacher closest to Teacher III, which is Teacher I, as the first teacher.

Our algorithm for designing teacher orders is lightweight. In fact, the main computation overhead of our algorithm is to optimize a set of tiny linear mappings ($\mathbb{R}^{256} \mapsto \mathbb{R}^{256}$ for FPN-based detectors). It takes about 3 GPU hours for each student model, which is negligible compared to the distillation process that takes hundreds of GPU hours.

Since our progressive knowledge distillation is a meta-level strategy, it can be combined with previous designs of distillation mechanisms, without much efforts. Starting with a student detector and a pool of candidate teachers, we can first select a subset of teachers and design their distillation order. In place of the simple feature matching loss, we then apply a more advanced distillation mechanism with each teacher sequentially to train the student detector.

## 4 EXPERIMENTS

We study the efficacy of our proposed strategy, progressive knowledge distillation, from multiple perspectives. First of all in Section 4.1, we use a controlled experiment to demonstrate that our heuristic Algorithm 1 consistently produces teacher orders that are near-optimal compared to all possibilities. Then in Section 4.2 and 4.3, we apply the progressive distillation strategy along with the simple feature-matching loss (Section 3.1) to show this strategy alone brings significant gains to knowledge distillation. Since our contribution of progressive distillation is orthogonal to previous efforts in designing distillation mechanisms, in Section 4.4 we then combine it with state-of-the-art distillation mechanisms to maximize the student performance, and we show our progressive distillation is the key to the success of distillation from Transformer-based teachers to convolution-based students. Finally in Section 4.5, we try to understand the performance gain of progressive distillation by analyzing the training loss dynamics.

**Student and teacher models:** To investigate the impact of different teacher models and their combinations, as shown in Table 1, we construct a variety of teacher-student pairs from a set of widely-used object detection and instance segmentation networks, including RetinaNet (Lin et al., 2017b), Mask R-CNN (He et al., 2017), FCOS (Tian et al., 2019), HTC (Chen et al., 2019a), and DetectoRS (Qiao et al., 2021). They have a wide range of runtime and detection performance. We select ResNet-50 backboned RetinaNet and Mask R-CNN as the student models (Student I & II), due to their low latency, simple structure, and wide application, for single-stage and two-stage object detection respectively. More advanced models such as DetectoRS have better detection performance, but require much more training/inference time, so we use them as teachers. We mainly consider the RetinaNet and Mask R-CNN as the student models, and lightweight variants with a smaller backbone (Student III), or reduced input resolution (Student IV).

**Datasets and evaluation metrics:** We mainly evaluate on the challenging object detection dataset MS COCO 2017 (Lin et al., 2014), which contains bounding boxes and instance segmentations for 80 common object categories. We train our models on the split of train2017 (118k images) and report results on val2017 (5k images). We report the standard COCO-style Average Precision (AP) metric and end-to-end latency (from images to predictions) as the runtime. We also evaluate

| $k$ | Suggested teacher order | Student AP | All student AP range | Ranking in all orders |
|---|---|---|---|---|
| 1 | IV | 36.7 | [36.2, 36.8] | 2 / 4 |
| 2 | III→IV | 37.6 | [36.2, 37.6] | 1 / 16 |
| 3 | I→III→IV | 37.9 | [36.2, 38.0] | 2 / 40 |
| 4 | I→III→IV | 37.9 | [36.2, 38.2] | 7 / 64 |

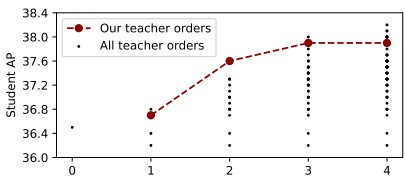

**Table 2: Comparison of teacher order suggested by Algorithm 1 with all other orders under limited training budgets (Li et al., 2020b).** $k$ denotes the maximum number of used teachers. **Left**: We show some statistics of possible student AP performance and the ranking of the student using our distillation order. **Right**: We visualize the comparative advantage of our teacher orders (red dots) over all other orders (black dots). Some black scatter points overlap due to the same student AP. Our proposed Algorithm 1 can consistently produce highly competitive distillation orders of teachers.

on another object detection dataset Argoverse-HD (Chang et al., 2019), and a more challenging evaluation protocol streaming perception (Li et al., 2020a). These results are in Appendix D.

**Baselines:** Our main contribution is *orthogonal* to previous methods: We leverage a sequence of teachers to distill the student, instead of designing a sophisticated distillation loss to better transfer knowledge from one single teacher. Since we are studying a new setting where multiple teachers are available, which is missing in previous literature, we mainly focus on the *absolute improvements* – the performance of our distilled student models compared with the original student models and with the performance upper-bound of the teacher models. We find using a sequence of teachers, instead of their ensemble, is more effective. Due to limited space, we leave this comparison in Appendix B&C.

### 4.1 SEARCHING FOR THE NEAR-OPTIMAL TEACHER ORDER

As we have discussed in Section 3.2, finding the optimal order of teachers for the progressive knowledge distillation takes factorial time complexity. To acquire a near-optimal teacher order, we propose the heuristic Algorithm 1. In this section, we will validate that this algorithm is near-optimal. To achieve this comprehensive comparison, we distill Student I with *all orders* of teachers from the pool Teacher I-IV. We use a reduced training budget: For each teacher, we only train the student for 3 epochs on MS COCO. We use the linear learning rate schedule, which has been shown comparably effective in a limited budget setting by Li et al. (2020b).

We first measure the adaptation costs among the student and teacher models. A visualization of the cost graph is shown in Figure 3. Following Algorithm 1, we can construct a sequence of teachers. We compare the teacher orders given by our proposed algorithm against *all other* orders, via the performance of the distilled student's performance. As shown in Table 2, teacher orders suggested by Algorithm 1 are consistently near-optimal in this setting. In the following sections, we will use order provided by Algorithm 1, without brute-force iterating over all possible orders. One may question that the greedy path selection shown in Figure 3 is be inferior to a global optimization algorithm. However, we find the later teachers impact the student performance more profoundly, so we need to greedily select teachers from the sequence tail. More details and comparison with other heuristics are provided in Appendix A.

We start by distilling RetinaNet and Mask R-CNN with a ResNet-50 backbone (Student I & II). Here we consider *homogeneous* teachers where the numbers of channels and the spatial resolutions of feature maps are *consistent* between the student and teacher. For the RetinaNet student, we still consider the pool of Teacher I-IV, the same as Section 4.1. For the Mask R-CNN student, we should no longer use Teacher I (the student itself) or Teacher II (the single-stage teacher does not outperform the student by a large margin). To compensate for that, we include Teacher V, which can be considered as a hybrid model of DetectoRS backbone/neck and Mask R-CNN head. Thus, the teacher pool for Mask R-CNN includes Teacher III-V. To control the total training time, we limit the number of teachers to be 2. Thus, we initialize from an off-the-shelf ('OTS') student, and sequentially distill it with 2 teachers, in total 24 epochs (equivalent to a $2\times$ training schedule). Besides the OTS students, we also compare with two other baselines: 1) students trained with a longer $3\times$ training schedule, and 2) students *directly distilled* by the final target teacher, using a $2\times$ training schedule. More architectural details are listed in Table 1.

Following Section 4.1, we use Algorithm 1 to determine the sequence of teachers to use for each student. For RetinaNet student, our algorithm suggests teacher sequence III→IV. For Mask R-

| ID | Model | Method | Box | | | | | | Mask | | | | | |
|----|-------|--------|-----|------|------|------|------|------|------|------|------|------|------|------|
| | | | AP | $AP_{50}$ | $AP_{75}$ | $AP_S$ | $AP_M$ | $AP_L$ | AP | $AP_{50}$ | $AP_{75}$ | $AP_S$ | $AP_M$ | $AP_L$ |
| 1 | | OTS | 36.5 | 55.4 | 39.1 | 20.4 | 40.3 | 48.1 | - | - | - | - | - | - |
| 2 | RetinaNet | Longer 3× training schedule | 39.5 | 58.8 | 42.2 | **23.8** | 43.2 | 50.3 | - | - | - | - | - | - |
| 3 | (Student I) | Directly distilled by Teacher IV | 39.5 | 58.6 | 41.9 | 21.0 | 42.8 | 54.0 | - | - | - | - | - | - |
| 4 | | **Progressively distilled by Teachers III→IV** | **39.9** | **59.2** | **42.7** | 21.7 | **43.3** | **54.1** | - | - | - | - | - | - |
| 5 | | OTS | 38.2 | 58.8 | 41.4 | 21.9 | 40.9 | 49.5 | 34.7 | 55.7 | 37.2 | 18.3 | 37.4 | 47.2 |
| 6 | Mask R-CNN | Longer 3× training schedule | 40.9 | 61.3 | 44.8 | **24.4** | 44.6 | 52.3 | 37.1 | 58.3 | **39.9** | 18.4 | 39.8 | 51.9 |
| 7 | (Student II) | Directly distilled by Teacher IV | 41.0 | 61.6 | 45.0 | 23.5 | 44.5 | 54.0 | 37.0 | 58.5 | 39.8 | 17.5 | 39.9 | 51.3 |
| 8 | | **Progressively distilled by Teachers V→IV** | **41.4** | **61.9** | **45.1** | 23.3 | **45.0** | **55.4** | **37.3** | **58.8** | 39.8 | **19.4** | **40.4** | **52.1** |

**Table 3: Homogeneous distillation of COCO detectors,** where students with ResNet-50 backbones are distilled with teachers with ResNet-50 backbones. We report the detection ('Box') and segmentation ('Mask') APs, and we compare our distilled student with off-the-shelf ('OTS') student, longer trained student, and the state-of-the-art distillation baselines. Our distilled student significantly improves the detection AP over the 'OTS' student by **3**.**4**% for RetinaNet and **3**.**2**% for Mask R-CNN, and outperforms the baselines.

CNN student, our algorithm suggests teacher sequence V→IV. Table 3 shows the results on COCO. Additional results, analysis, and ablation studies of Mask R-CNN distillation are in Appendix B.

## 4.2 DISTILLATION WITH HOMOGENEOUS TEACHERS

**Overall performance:** Our distilled student models (row 4&8) significantly improves over the 'OTS' students (row 1&5). The box AP of RetinaNet is improved from 36.5% to 39.9% (+3.4%). The box AP of Mask R-CNN is improved from 38.2% to 41.4% (+3.2%) and the mask AP of Mask R-CNN is improved from 34.7% to 37.3% (+2.6%). After progressive distillation, our resulting Mask R-CNN detector has *comparable performance with HTC teacher, but much less runtime* (51ms vs. 181ms).

**Comparison with baselines:** First, the performance gain is not merely from a longer training schedule. Our distilled student models (row 4&8) consistently outperform original students trained with a 3× schedule (row 2&6). Second, progressive distillation using a curriculum of teachers (row 4&8) is better than direct distillation from a strong teacher (row 3&7), even if the total training time is the same. It is worth noting that our detection performance for large objects receives the most gain (about 6% $AP_L$ improvement for both models). The reason why we emphasize $AP_L$ is that, in an efficiency-centric real-world application (e.g. autonomous driving, robot navigation), detecting nearby larger objects is more crucial than others. From a realistic perspective, better $AP_L$ shows better applicability of our approach.

## 4.3 DISTILLATION WITH HETEROGENEOUS TEACHERS

To validate our progressive distillation approach is general, we now consider a more challenging heterogeneous scenario, where students and teachers have different backbones or input resolutions. Specifically, Student III, a ResNet-18 Mask R-CNN, is distilled with ResNet-50 teachers; Student IV, a model with reduced input resolution, is distilled with teachers trained with larger input resolutions. The results are summarized in Table 4, and additional results are included in Appendix C.

**Heterogeneous backbones:** Student III has a ResNet-18 backbone and about half runtime as its ResNet-50 counterpart (Teacher I). We find the proper distillation scheme for Student III is to use the sequence of Teacher I→V→IV, which significantly improves Student III over the 'OTS' model. The box AP of Student III is improved from 33.3% to 37.0% (+3.7%), and especially for large objects, $AP_L$ is improved from 43.6% to 50.0% (+6.4%).

**Heterogeneous input resolutions:** Although inputs with varying resolutions can be fed into most object detectors without changing the architecture, the performance often degenerates when there is a resolution mismatch between training and evaluation (Tan et al., 2020; Li et al., 2020a). If ultimately we want to apply a detector to low-resolution inputs for fast inference, it is better to use low-resolution inputs during training. On the other hand, we conjecture that teachers with high-resolution inputs may provide finer details that can assist the student. With our progressive distillation approach, we investigate the improvement of a low-resolution student distilled by a sequence of teachers with high-resolution inputs. We denote the standard input resolution $1333 \times 800$ as $1\times$, and a reduced resolution $333 \times 200$ as $0.25\times$. We distill Student IV (with $0.25\times$ resolution) by a sequence of Teacher I variants ($0.5\times \rightarrow 0.75\times \rightarrow 1\times$). From Table 4, we can see substantial improvement brought by progressive knowledge distillation: the box AP is improved from 25.8% to 31.5% (+5.7%) and the mask AP is improved from 23.0% to 28.2% (+5.2%).

| ID | Model | Backbone | Resolution | AP Box | AP Mask |
|----|-------|----------|------------|--------|---------|
| 1 | Student III, OTS | R18 | 1× | 33.3 | 30.5 |
| 2 | Student III, **Our distilled** | R18 | 1× | **37.0** | **33.7** |
| 3 | Student IV, OTS | R50 | 0.25× | 25.8 | 23.0 |
| 4 | Student IV, **Our distilled** | R50 | 0.25× | **31.5** | **28.2** |

Table 4: **Heterogeneous distillation of COCO detectors**, where students with smaller backbones (ResNet-18 vs. ResNet-50) or input resolutions ($333 \times 200$ vs. $1333 \times 800$) are distilled with heterogeneous teachers, requiring additional transfer logic (Sec. 3.1). We report the detection ('Box'), segmentation ('Mask') APs and runtime, and compare our distilled student with its teachers (see Table 1) and off-the-shelf ('OTS') student. Our progressive distillation significantly improves the 'OTS' students by over **3**% AP.

| ID | Model | Distillation | AP Box | AP Mask |
|----|-------|--------------|--------|---------|
| 1 | RetinaNet | Direct RetinaNet/Swin-S | 41.0 | - |
| 2 | (Student I) | **Progressive RetinaNet/Swin-T→S** | **42.0** | - |
| 3 | Mask R-CNN | Direct MRCNN/Swin-S | 42.0 | 37.7 |
| 4 | (Student II) | **Progressive MRCNN/Swin-T→S** | **42.5** | **38.4** |

Table 5: **Distillation from Transformer-based teachers** (Liu et al., 2021) **to convolution-based students.** Due to the architectural difference and capacity gap, directly distilling from a stronger teacher with Swin-S backbone does not yield better students than convolution-based teachers in Figure 4. An intermediate Swin-T teacher and *progressive distillation* solve this issue without increasing training time. Compared to off-the-shelf models, our RetinaNet and Mask R-CNN students improve by **5.5**% AP and **4.3**% box AP, respectively.

## 4.4 COMBINATION WITH STATE-OF-THE-ART DISTILLATION MECHANISMS

Our meta-level strategy of using a sequence of teachers to progressively distill a student is independent of choice of the distillation mechanism for each teacher. We have shown progressive distillation can boost the simple distillation based on feature matching above, and in this section, we will combine progressive distillation with state-of-the-art distillation mechanisms for object detection to further improve student accuracy.

**Distillation protocol:** We evaluate our progressive distillation with three most recent works on detector distillation: CWD (Shu et al., 2021), FGD (Yang et al., 2022a), and MGD (Yang et al., 2022b). To ensure fair comparison, we use the *same teacher-student pairs* as them: RetinaNet/ResNet-50 and RetinaNet/ResNeXt-101 are the single-stage student and final teacher. Mask R-CNN/ResNet-50 and Cascade Mask R-CNN/ResNeXt-101-DCN are the two-stage student and final teacher. Between them, we insert one medium-capacity teacher to progressively distill the student: RetinaNet/ResNet-101 for single-stage and Cascade Mask R-CNN/ResNet50-DCN for two-stage. Also for fairness, we *keep the total training epochs the same*. We set "1×" training schedule for each teacher, so that the total training time is equivalent to "2×", the same as in previous works.

Figure 4 shows our progressive distillation strategy consistently improves students' final accuracy. For example, the performance of FGD-distilled RetinaNet/ResNet-50 improves from $40.7\%$ to $41.5\%$ AP ($+0.8\%$), and this gain is larger than mechanism advance from FGD to MGD ($+0.3\%$). We bring performance gains to state-of-the-art detection distillation almost *for free*.

Next, we investigate how to further maximize the student performance. Swin Transformer (Liu et al., 2021) can act as an even stronger teacher than the convolution-based teachers used in previous works. However compared to convolution-based teachers, directly distill from such a teacher cannot improve the student performance, even if we use the state-of-the-art method MGD. For example, RetinaNet/Swin-Small ($47.1\%$ AP) is much stronger than RetinaNet/ResNeXt-101($41.6\%$ AP), but direct distillation from both yields the same student performance ($41.0\%$ AP). To bridge the architectural difference and capacity gap between the ResNet-50 student and Swin-Small teacher, we can utilize an intermediate Swin-Tiny teacher. As shown in Table 5, this progressive approach brings the best students: the performance of ResNet-50 based RetinaNet increases to $42.0\%$ AP and Mask R-CNN increases to $42.5\%$ AP.

## 4.5 UNPACKING THE PERFORMANCE GAIN: GENERALIZATION OR OPTIMIZATION?

We have shown that our distilled student significantly improves the accuracy on the *validation* data over the off-the-shelf student. As further demonstrated in Figure 5a, the validation accuracy of the distilled student gradually increases during distillation, and achieves a higher value compared with the student trained without teachers. A natural question then arises – why is distillation helping? There are two possible hypotheses: (1) *improved optimization:* distillation facilitates the optimization procedure, leading to a better local minimum, and (2) *improved generalization:* the distillation process helps the student generalize to unseen data.

Improved optimization is typically manifested through a better model, a lower training loss and a higher validation accuracy, which is exactly the case for Mask R-CNN, HTC and DetectoRS.

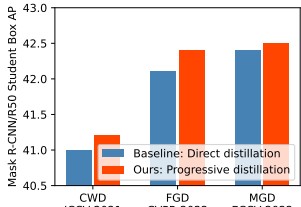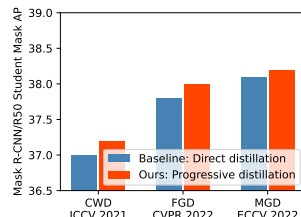

**Figure 4: Our progressive distillation strategy consistently benefits state-of-the-art distillation mechanisms.** Using an intermediate RetinaNet/ResNet-101 teacher between RetinaNet/ResNet-50 student and RetinaNet/ResNeXt-101 teacher (**Left**), or Cascade Mask R-CNN/ResNet50-DCN between Mask R-CNN/ResNet-50 and Cascade Mask R-CNN/ResNeXt-101-DCN (**Middle** for Box AP and **Right** for Mask AP), we improve the direct distillation baselines by 0.2% to 0.8% AP, *without increasing training time*.

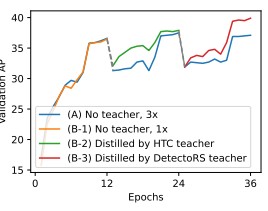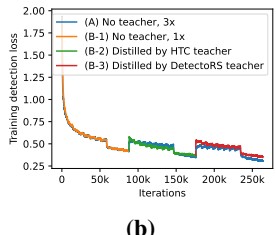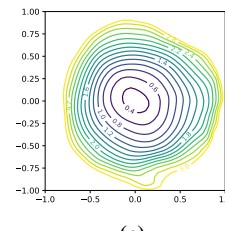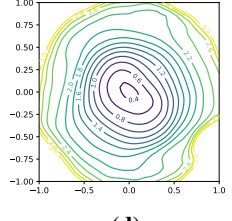

| (a) | (b) | (c) | (d) |

**Figure 5: Comparisons of student models trained with and without teachers.** We train a ResNet-50 backboned RetinaNet (Student I) with: (A) a prolonged 3× training schedule (curves in blue); (B) progressive knowledge distillation from HTC (Teacher III) and then DetectoRS (Teacher IV) (curves in orange-green-red). We compare the validation AP (Figure 5a) and the training detection loss $L_{detect}$ (Figure 5b) of the two students during the training process. Despite its worse training loss, the distilled student can generalize better on the validation set. We also compare the loss landscapes (Li et al., 2018) of the original student (Figure 5c) and the distilled student (Figure 5d). Distillation can guide the student to converge to a flatter local minimum. These observations suggest distillation *helps generalization rather than optimization*.

Consequently, one might think that distillation works in the same way. However, our investigation suggests the opposite – our progressive distillation increases both the validation accuracy and the training loss, and therefore effectively reduces the generalization gap. In Figure 5, we compare the original RetinaNet model and the distilled student, which have the same architecture, the same latency and are trained on the same data, but with different supervision (only ground-truth labels vs. additional knowledge distillation). To eliminate the influence of learning rate changes, we train the original student with a 3× schedule and restart the learning rate at the same time with the distilled student. Interestingly, although distillation can improve the student's validation performance, the *training* detection loss of the distilled student is higher than the original student. This suggests that distillation does *not* help the optimization process to find a local minimum with a lower training loss, but rather strengthen the generalizability of the student model.

To further support this observation, we also visualize the local loss landscape (Li et al., 2018). The distilled student has a flatter loss landscape (Figure 5d) compared to the original one (Figure 5c). As widely believed in the machine learning literature, flat minima lead to better generalization (Hochreiter & Schmidhuber, 1997; Keskar et al., 2017). The observation shown in Figure 5 is illustrated for RetinaNet, but we also have similar observation in other students. As a conclusion, knowledge distillation, which enforces the student to mimic the teachers' features, can be considered as an implicit regularization, and helps the student combat overfitting and achieve better generalization.

## 5 CONCLUSION

We present a simple yet effective approach to knowledge distillation, which progressively transfers the knowledge of a sequence of teachers to learn a lightweight object detector. Our approach automatically arranges multiple teachers into a curriculum, and thus effectively mitigating the capacity gap between the teacher and student. We successfully distill knowledge from Transformer-based teachers to convolution-based students, and achieve state-of-the-art performance on the challenging COCO dataset. Our analysis also finds distillation improves generalization rather than optimization.

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

APPENDIX

We summarize the content of this supplementary document as follows. Section A includes additional results and analysis of our proposed algorithm for teacher order selection. Sections B and C provide additional ablation study on distillation with homogeneous teachers and heterogeneous teachers, respectively. Section D shows the generalizability of our approach, which demonstrate the experimental results of the Argoverse-HD dataset with streaming accuracy metric. Section E compares our work with prior knowledge distillation methods in detail. Section F provides some additional experiments on combining our method with state-of-the-art distillation mechanism. Section G lists our implementation details.

## A    MORE RESULTS ON SEARCHING FOR THE NEAR-OPTIMAL TEACHER ORDER

In this section, we show more detailed results about searching a proper teacher order for progressive knowledge distillation, and validate the approach we propose in the main paper. As described in Section 3.2, we first quantify the adaptation cost $\mathcal{C}(\cdot, \cdot)$ between every pair of models in our pool, and then use a heuristic method (Algorithm 1) to construct a sequence of teachers. We have shown that the teacher order suggested by our algorithm is highly competitive in Table 2. One might think there should be better choices than a greedy algorithm on a directed graph, such as a shortest-path algorithm. To validate our algorithm design, we compare our Algorithm 1 against several other algorithms.

---

**Algorithm 1:** Determining the Teacher Order

**Input:** Student model $S$, pool of teacher models $\mathcal{P} = \{T_i\}_{i=1}^N$, teacher models' performance $\{Q(T_i)\}_{i=1}^N$, maximum number of selected teachers $k$
**Output:** Sequence of teachers $\mathcal{O}, \text{len}(\mathcal{O}) \le k$

1 Pick the best performing teacher: $T_{\alpha_k} \leftarrow \arg\max_{T_u \in \mathcal{P}} Q(T_u), \mathcal{O} \leftarrow [T_{\alpha_k}]$
2 Exclude from pool: $\mathcal{P} \leftarrow \mathcal{P} \setminus \{T_{\alpha_k}\}$
3 **for** $j \leftarrow k-1$ **to** 1 **do**
4      Get candidate sub-pool: $\mathcal{P}_j = \{T_u \mid T_u \in \mathcal{P}, \mathcal{C}(T_u, T_{\alpha_{j+1}}) < \mathcal{C}(S, T_{\alpha_{j+1}})\}$
5      **if** $\mathcal{P}_j \neq \emptyset$ **then**
6          Pick the teacher closest to $T_{\alpha_{j+1}}$: $T_{\alpha_j} \leftarrow \arg\min_{T_u \in \mathcal{P}_j} \mathcal{C}(T_u, T_{\alpha_{j+1}})$
7          Prepend $T_{\alpha_j}$ to $\mathcal{O}$
8          Exclude from pool: $\mathcal{P} \leftarrow \mathcal{P} \setminus \{T_{\alpha_j}\}$
9      **else** Break
10 **return** $\mathcal{O}$

---

To begin with, we include the detailed adaptation costs $\mathcal{C}(\cdot, \cdot)$ among RetinaNet (Student I) and its teachers (Teacher I-IV) in Table 6. As described in Section 4.1, we have distilled Student I with *all* possible teacher orders in the pool, using a reduced training budget of 3 epochs for each teacher. The results of these mini-budget distillation are summarized in Table 7.

**Table 6:** Adaptation costs among Student I (RetinaNet) and Teacher I-IV (Mask R-CNN, FCOS, HTC, DetectoRS). The adaptation cost is computed pair-wise as described in Section 3.2 of the main paper. Using this metric we can construct a directed graph, as illustrated in Figure 3.

| From \ To | Student I | Teacher I | Teacher II | Teacher III | Teacher IV |
|---|---|---|---|---|---|
| Student I | - | 0.939 | 0.060 | 1.568 | 1.254 |
| Teacher I | 0.183 | - | 0.070 | 0.934 | 0.963 |
| Teacher II | 0.339 | 1.181 | - | 1.940 | 1.401 |
| Teacher III | 0.191 | 0.484 | 0.082 | - | 0.890 |
| Teacher IV | 0.232 | 0.767 | 0.077 | 1.248 | - |

**Table 7:** Performance of Student I (RetinaNet) distilled with different teacher sequences, under reduced training budgets. For each teacher in the sequence, the student is trained for 3 epochs on COCO. After progressive knowledge distillation, the student is evaluated on the COCO validation set. The teacher orders suggested by Algorithm 1 are marked **bold**.

| Length | Teacher Sequence | Student AP | Length | Teacher Sequence | Student AP | Length | Teacher Sequence | Student AP |
|---|---|---|---|---|---|---|---|---|
| 1 | III | 36.8 | | III→II→IV | 38.0 | | III→II→I→IV | 38.2 |
| | **IV** | **36.7** | | **I→III→IV** | **37.9** | | III→I→II→IV | 38.1 |
| | I | 36.4 | | III→IV→II | 37.9 | | I→III→II→IV | 38.1 |
| | II | 36.2 | | II→III→IV | 37.9 | | II→III→I→IV | 38.0 |
| | | | | III→I→IV | 37.8 | | I→III→IV→II | 38.0 |
| 2 | **III→IV** | **37.6** | | I→II→IV | 37.7 | | II→I→III→IV | 37.9 |
| | IV→II | 37.3 | | I→IV→II | 37.6 | | III→I→IV→II | 37.9 |
| | III→II | 37.3 | | IV→II→III | 37.5 | | I→II→III→IV | 37.9 |
| | I→IV | 37.3 | | IV→III→II | 37.5 | | IV→I→III→II | 37.7 |
| | IV→III | 37.2 | | II→I→IV | 37.5 | | II→I→IV→III | 37.7 |
| | I→III | 37.1 | | I→III→II | 37.5 | | I→II→IV→III | 37.7 |
| | IV→I | 37.0 | | IV→I→III | 37.4 | | IV→III→I→II | 37.6 |
| | II→IV | 37.0 | 3 | II→IV→III | 37.4 | 4 | IV→I→II→III | 37.6 |
| | III→I | 37.0 | | III→IV→I | 37.4 | | III→IV→II→I | 37.6 |
| | II→I | 36.9 | | III→I→II | 37.4 | | III→IV→I→II | 37.6 |
| | II→III | 36.8 | | I→IV→III | 37.4 | | III→II→IV→I | 37.6 |
| | I→II | 36.8 | | IV→II→I | 37.3 | | I→IV→III→II | 37.6 |
| | | | | IV→III→I | 37.3 | | IV→II→I→III | 37.5 |
| | | | | IV→I→II | 37.3 | | IV→III→II→I | 37.5 |
| | | | | I→II→III | 37.3 | | II→IV→I→III | 37.5 |
| | | | | II→IV→I | 37.2 | | II→III→IV→I | 37.5 |
| | | | | II→I→III | 37.2 | | I→IV→II→III | 37.5 |
| | | | | III→II→I | 37.2 | | II→IV→III→I | 37.4 |
| | | | | II→III→I | 37.1 | | IV→II→III→I | 37.3 |

**Table 8:** Comparison of four algorithms for teacher order selection, in the mini-budget distillation setting. Our Algorithm 1 can consistently produce a better teacher order than other algorithms.

| $k$ | Algorithm | Suggested teacher order | Student AP | Ranking in all orders | $k$ | Algorithm | Suggested teacher order | Student AP | Ranking in all orders |
|---|---|---|---|---|---|---|---|---|---|
| 1 | Shortest-path (sum) | IV | 36.7 | 2 / 4 | 3 | Shortest-path (sum) | II→I→IV | 37.5 | 9 / 40 |
| | Shortest-path (max) | IV | 36.7 | 2 / 4 | | Shortest-path (max) | I→III→IV | 37.9 | 2 / 40 |
| | Forward construction | II | 36.2 | 4 / 4 | | Forward construction | II→I→III | 37.2 | 25 / 40 |
| | Our Algorithm 1 | IV | 36.7 | 2 / 4 | | Our Algorithm 1 | I→III→IV | 37.9 | 2 / 40 |
| 2 | Shortest-path (sum) | II→IV | 37.0 | 7 / 16 | 4 | Shortest-path (sum) | II→I→III→IV | 37.9 | 7 / 64 |
| | Shortest-path (max) | I→IV | 37.3 | 2 / 16 | | Shortest-path (max) | II→I→III→IV | 37.9 | 7 / 64 |
| | Forward construction | II→I | 36.9 | 10 / 16 | | Forward construction | II→I→III→IV | 37.9 | 7 / 64 |
| | Our Algorithm 1 | III→IV | 37.6 | 1 / 16 | | Our Algorithm 1 | I→III→IV | 37.9 | 7 / 64 |

Given the adaptation costs in Table 6, we can construct a directed graph, part of which has been illustrated in Figure 3. On the directed graph, we can run several algorithms to select a path. Besides our Algorithm 1, one may also propose these algorithms:

- **Shortest-path (sum)**: Set the student as the source node, and set the best performing teacher as the target node $T_{\lambda_k}$. Find a path $S \to T_{\lambda_1} \to \cdots \to T_{\lambda_k}$ that minimizes the *sum* of adaptation costs along the path:
  $\min_{T_{\lambda_1},\ldots,T_{\lambda_{k-1}}} \mathcal{C}(S, T_{\lambda_1}) + \sum_{j=1}^{k-1} \mathcal{C}(T_{\lambda_j}, T_{\lambda_{j+1}})$.
- **Shortest-path (max)**: Set the student as the source node, and set the best performing teacher as the target node $T_{\lambda_k}$. Find a path $S \to T_{\lambda_1} \to \cdots \to T_{\lambda_k}$ that minimizes the *maximum* of adaptation costs along the path: $\min_{T_{\lambda_1},\ldots,T_{\lambda_{k-1}}} \max\{\mathcal{C}(S, T_{\lambda_1}), \mathcal{C}(T_{\lambda_1}, T_{\lambda_2}), \ldots, \mathcal{C}(T_{\lambda_{k-1}}, T_{\lambda_k})\}$.
- **Forward construction**: Contrary to Algorithm 1, we may start from the student and choose the nearest teacher from the current one, to construct the sequence:
  $T_{\lambda_1} \leftarrow \arg\min_{T_u \in \mathcal{P}} \mathcal{C}(S, T_u), T_{\lambda_{j+1}} \leftarrow \arg\min_{T_u \in \mathcal{P}} \mathcal{C}(T_{\lambda_j}, T_u)$.

The output teacher sequences and corresponding student performance of these three algorithms are summarized in Table 8. In this setting, our Algorithm 1 can consistently produce a competitive teacher order that leads to a good performance of the distilled student. Compared to our Algorithm 1, shortest-path (max) can achieve a similar performance, and it is only worse than ours when $k = 2$. Forward construction performs worst among the four algorithms.

In summary, a greedy backward construction like Algorithm 1 works the best in our setting, rather than globally optimized shortest-path algorithms. The final target teacher has the most profound impact on the distilled student's performance. In order to fully assist the final teacher, we need to use another teacher with the minimal adaptation cost to the final teacher before it, which is exactly the behavior of Algorithm 1.

## B   ABLATION STUDY ON DISTILLATION WITH HOMOGENEOUS TEACHERS

In this section, we provide more details about distillation with homogeneous teachers (Section 4.2). We investigate (1) the impact of each individual teacher; and (2) distillation with teachers simultaneously vs. sequentially.

**Table 9:** Homogeneous distillation of COCO detectors, where students with ResNet-50 backbones are distilled with teachers with ResNet-50 backbones. We report the detection ('Box') and segmentation ('Mask') APs and runtime, and we compare our distilled student with its teachers, off-the-shelf ('OTS') student. Our distilled student significantly improves the APs over the 'OTS' student by around 3%.

| ID | Model | Box | | | | | | Mask | | | | | | Runtime |
|---|---|---|---|---|---|---|---|---|---|---|---|---|---|---|
| | | AP | $AP_{50}$ | $AP_{75}$ | $AP_S$ | $AP_M$ | $AP_L$ | AP | $AP_{50}$ | $AP_{75}$ | $AP_S$ | $AP_M$ | $AP_L$ | (ms) |
| 1 | Teacher III | 42.3 | 61.1 | 45.8 | 23.7 | 45.6 | 56.3 | 37.4 | 58.4 | 40.2 | 19.6 | 40.4 | 51.7 | 181 |
| 2 | Teacher IV | 49.1 | 67.7 | 53.4 | 29.9 | 53.0 | 65.2 | 42.6 | 65.1 | 46.0 | 24.1 | 46.4 | 58.6 | 223 |
| 3 | Teacher V | 45.1 | 66.3 | 49.3 | 27.8 | 49.0 | 59.3 | 40.1 | 63.1 | 42.8 | 22.9 | 43.8 | 54.8 | 142 |
| 4 | Student II (OTS) | 38.2 | 58.8 | 41.4 | 21.9 | 40.9 | 49.5 | 34.7 | 55.7 | 37.2 | 18.3 | 37.4 | 47.2 | 51 |
| 5 | Student II (distilled) | **41.4** | **61.9** | **45.1** | **23.3** | **45.0** | **55.4** | **37.3** | **58.8** | **39.8** | **19.4** | **40.4** | **52.1** | 49 |

**Table 10:** Ablation study of homogeneous distillation of COCO detectors (models in Table 9). Our distillation strategy is *consistently effective irrespective of teacher type*. Moreover, sequential distillation with two teachers outperforms both distillation with a single teacher and simultaneous distillation with two teachers. Our best distilled student is obtained by *progressive* distillation, where Student II is first distilled with Teacher V (a weaker, more similar teacher with the same head as Student II) and then distilled with Teacher IV (a stronger teacher whose architecture is completely different from Student II).

| ID | Student II | Box | | | | | | Mask | | | | | |
|---|---|---|---|---|---|---|---|---|---|---|---|---|---|
| | | AP | $AP_{50}$ | $AP_{75}$ | $AP_S$ | $AP_M$ | $AP_L$ | AP | $AP_{50}$ | $AP_{75}$ | $AP_S$ | $AP_M$ | $AP_L$ |
| 1 | OTS | 38.2 | 58.8 | 41.4 | 21.9 | 40.9 | 49.5 | 34.7 | 55.7 | 37.2 | 18.3 | 37.4 | 47.2 |
| 2 | Distilled by Teacher III | 40.2 | 60.7 | 43.8 | 22.5 | 43.8 | 53.4 | 36.3 | 57.3 | 38.7 | 18.9 | 39.3 | 50.3 |
| 3 | Distilled by Teacher IV | 40.8 | 61.5 | 44.6 | 23.0 | 44.3 | 54.2 | 36.8 | 58.3 | 39.4 | 19.2 | 39.9 | 51.0 |
| 4 | Distilled by Teacher V | 40.8 | 61.4 | 44.5 | 22.9 | 44.3 | 54.2 | 36.6 | 58.1 | 39.1 | 19.2 | 39.6 | 51.0 |
| 5 | Distilled by Teachers IV+V | 39.8 | 60.3 | 43.4 | 22.1 | 43.3 | 52.9 | 35.9 | 57.1 | 38.1 | 18.3 | 39.0 | 49.8 |
| 6 | Distilled by Teachers IV→V | 41.0 | 61.7 | 44. 8 | 23.0 | 44.3 | 54.9 | 36.8 | 58.3 | 39.2 | 19.5 | 39.9 | 51.3 |
| 7 | Distilled by Teachers V→IV | **41.4** | **61.9** | **45.1** | **23.3** | **45.0** | **55.4** | **37.3** | **58.8** | **39.8** | **19.4** | **40.4** | **52.1** |

**Impact of individual teachers:** We first distill Student II with each of the three teachers individually: Teacher III has the same backbone and neck but a more advanced head; Teacher IV has more advanced backbone, neck, and head; Teacher V has the same head but more advanced backbone and neck. Table 9 provides the performance of the three teachers, where Teacher IV achieves the best performance (row 1-3). From Table 10, we can see that our distilled students (row 2-7) *significantly and consistently* outperform the off-the-shelf student (row 1), demonstrating the effectiveness of our distillation strategy *irrespective of the types of teachers*. Moreover, the improvement of the student distilled with Teacher V (row 2) over that with Teacher III (row 3) shows that a more powerful teacher generally leads to a better distilled student. Interestingly, although Teacher IV is more powerful than Teacher V, Table 10 shows that their distilled students achieve quite similar AP (row 2 vs. row 4). This indicates that an even more powerful teacher does not necessarily further improve the

distilled student; too large a capacity and structure gap between the teacher and student will limit the effectiveness of distillation. Also, it is easier to distill from teachers with the same head.

**Simultaneous vs. progressive distillation:** We now distill Student II with the combined teachers, and we choose the top-performing Teacher IV and Teacher V. We investigate two types of combination – simultaneous distillation with a feature matching loss between each teacher and the student (row 5), and sequential distillation with teachers one by one (row 6-7). First, we find that using both teachers simultaneously (row 5) is *even worse* than our method using a single teacher (row 2-4). This shows that integrating different types of knowledge from multiple teachers is not a trivial task – simultaneously using the features from multiple teachers might provide *conflicting supervisions* to the student model and thus hinder its distillation process. By contrast, our sequential distillation overcomes this issue and improves the performance *irrespective of the order of the teachers* (row 6-7 vs. row 1-4). Second, the sequential order of the teachers makes a difference. A *curriculum-like progression* (row 7), where the teacher with a smaller adaptation cost is used first and that with a larger adaptation cost & a higher performance is used later, leads to the best performance.

**Overall performance:** Our best distillation performance is achieved when we first distill Student II with a curriculum of teachers (Teacher V→IV). Overall, the box AP is improved from 38.2% to 41.4% and the mask AP is improved from 34.7% to 37.3%. Our resulting Mask R-CNN detector has *comparable performance with HTC, but much smaller runtime*.

## C  ABLATION STUDY ON DISTILLATION WITH HETEROGENEOUS TEACHERS

In this section, we provide more details about distillation with heterogeneous teachers (Section 11). We investigate the heterogeneous cases where the backbones or input resolutions are different between the teachers and student.

**Table 11:** Heterogenous distillation of COCO detectors, where students with ResNet-18 backbones are distilled with teachers with ResNet-50 backbone, requiring additional transfer logic. We report the detection ('Box') and segmentation ('Mask') APs and runtime, and we compare our distilled student with its teachers, and off-the-shelf ('OTS') student. Our distilled student significantly improves the APs over the 'OTS' student by over 3%.

| ID | Model | Box | | | | | | Mask | | | | | | Runtime |
|---|---|---|---|---|---|---|---|---|---|---|---|---|---|---|
| | | AP | $AP_{50}$ | $AP_{75}$ | $AP_S$ | $AP_M$ | $AP_L$ | AP | $AP_{50}$ | $AP_{75}$ | $AP_S$ | $AP_M$ | $AP_L$ | (ms) |
| 1 | Teacher I | 38.2 | 58.8 | 41.4 | 21.9 | 40.9 | 49.5 | 34.7 | 55.7 | 37.2 | 18.3 | 37.4 | 47.2 | 51 |
| 2 | Teacher III | 42.3 | 61.1 | 45.8 | 23.7 | 45.6 | 56.3 | 37.4 | 58.4 | 40.2 | 19.6 | 40.4 | 51.7 | 181 |
| 3 | Teacher IV | 49.1 | 67.7 | 53.4 | 29.9 | 53.0 | 65.2 | 42.6 | 65.1 | 46.0 | 24.1 | 46.4 | 58.6 | 223 |
| 4 | Teacher V | 45.1 | 66.3 | 49.3 | 27.8 | 49.0 | 59.3 | 40.1 | 63.1 | 42.8 | 22.9 | 43.8 | 54.8 | 142 |
| 5 | Student III (OTS) | 33.3 | 52.9 | 35.9 | 18.2 | 35.9 | 43.6 | 30.5 | 50.0 | 32.1 | 15.5 | 32.9 | 41.8 | 29 |
| 6 | Student III (Distilled) | **37.0** | **56.8** | **39.9** | **20.2** | **39.8** | **50.0** | **33.7** | **53.6** | **36.0** | **17.2** | **36.0** | **47.3** | 29 |

**Overall performance:** Again, Tables 11 and 12 show that our distillation strategy is consistently effective with respect to all the teachers and their combinations, *e.g.*, the box AP improves from 33.3% to 37.0% and the mask AP improves from 30.5% to 33.7%.

**Two signature findings in heterogeneous distillation:** Compared to the homogeneous case, we find the capacity gap between models is a more important factor, and to bridge this gap a proper teacher order plays a more critical role. Details are explained as follows.

*The student-teacher capacity gap is more pronounced in heterogeneous distillation.* Among the four teachers, Teacher I shares exactly the same neck and head structure with the student, and has a similar but larger backbone; Teacher V has the same head with the student as well, but has a different backbone and neck; Teacher III has similar backbone and neck, but has a different head; and Teacher IV is the most powerful one with completely different architecture. Table 12 (rows 3-6) summarizes the distillation results with single teachers. First, directly distilling from the strongest teacher (Teacher IV) does not yield the largest improvement. Second, a relatively less powerful but more similar teacher (Teacher I) leads to the best distillation performance, improving the APs by 2%, although teachers V, III, and IV are all stronger than Teacher I. One possible reason is that Teacher I has the same neck and head as Student III as well as similar but deeper backbone, so the capacity gap between Student III and Teacher I is the smallest. Finally, we find that Teacher III is a strong but not particularly helpful teacher, achieving the worst distillation results. One possible reason is that

**Table 12:** Ablation study for heterogeneous COCO detector distillation (models in Table 11). Student III (Mask R-CNN with a ResNet-18 backbone) is distilled with teachers with *different* and larger ResNet-50 backbones. Training Student III for more epochs improves its performance, but not as much as progressive distillation with teachers. Note that for each distillation we train 12 epochs. Our distillation strategy is *consistently effective irrespective of the types of teachers*. Moreover, our sequential distillation with multiple teachers outperforms simultaneous distillation with multiple teachers. Our best distilled student is obtained by *progressive* distillation, where Student III is first distilled with Teacher I (a most similar teacher with the same head and neck as Student III and a deeper backbone), then distilled with Teacher V (a stronger teacher with the same head as Student III), and finally distilled with Teacher IV (a strongest teacher whose architecture is completely different from Student III).

| ID | Model | Box | | | | | | Mask | | | | | |
|----|-------|-----|------|------|------|------|------|------|------|------|------|------|------|
| | | AP | $AP_{50}$ | $AP_{75}$ | $AP_S$ | $AP_M$ | $AP_L$ | AP | $AP_{50}$ | $AP_{75}$ | $AP_S$ | $AP_M$ | $AP_L$ |
| 1 | Student III (OTS) | 33.3 | 52.9 | 35.9 | 18.2 | 35.9 | 43.6 | 30.5 | 50.0 | 32.1 | 15.5 | 32.9 | 41.8 |
| 2 | +12 epochs | 34.6 | 54.5 | 37.2 | 18.8 | 36.9 | 46.1 | 31.6 | 51.5 | 33.6 | 15.8 | 33.7 | 44.0 |
| 3 | +24 epochs | 34.5 | 54.2 | 37.2 | 18.8 | 36.5 | 45.8 | 31.5 | 51.2 | 33.8 | 16.0 | 33.4 | 43.7 |
| 4 | +36 epochs | 34.6 | 54.2 | 37.4 | 18.6 | 36.9 | 46.7 | 31.6 | 51.1 | 33.8 | 15.7 | 33.6 | 44.3 |
| 3 | Distilled by Teacher I | 35.8 | 55.8 | 38.8 | 19.3 | 38.8 | 47.9 | 32.6 | 52.7 | 34.8 | 16.0 | 35.3 | 45.5 |
| 4 | Distilled by Teacher III | 35.2 | 55.2 | 37.8 | 19.1 | 37.8 | 47.4 | 32.1 | 52.0 | 34.0 | 16.1 | 34.5 | 45.2 |
| 5 | Distilled by Teacher IV | 35.5 | 55.2 | 38.2 | 19.0 | 37.9 | 48.0 | 32.4 | 51.9 | 34.5 | 15.9 | 34.8 | 45.6 |
| 6 | Distilled by Teacher V | 35.4 | 55.2 | 38.3 | 19.4 | 37.9 | 48.4 | 32.2 | 52.2 | 34.3 | 15.4 | 34.4 | 45.8 |
| 7 | Distilled by Teachers IV+V | 34.8 | 54.9 | 37.2 | 19.0 | 37.2 | 47.0 | 31.6 | 51.7 | 33.9 | 15.7 | 33.8 | 44.2 |
| 8 | Distilled by Teachers I+IV+V | 36.0 | 55.4 | 39.1 | 18.2 | 38.1 | 48.3 | 32.1 | 53.0 | 34.7 | 15.8 | 34.7 | 46.1 |
| 9 | Distilled by Teachers I+III+IV+V | 36.1 | 55.2 | 39.0 | 18.4 | 38.2 | 48.0 | 31.7 | 52.9 | 34.3 | 15.1 | 34.2 | 46.3 |
| 10 | Distilled by Teachers I→V | 36.5 | 56.3 | 39.3 | 19.5 | 38.8 | 49.4 | 33.2 | 53.2 | 35.3 | 16.4 | 35.4 | 46.8 |
| 11 | Distilled by Teachers V→IV | 35.2 | 55.2 | 37.8 | 19.1 | 37.8 | 47.4 | 32.1 | 52.0 | 34.0 | 16.1 | 34.5 | 45.2 |
| 12 | Distilled by Teachers I→V→IV | **37.0** | **56.8** | **39.9** | **20.2** | **39.8** | **50.0** | **33.7** | **53.6** | **36.0** | **17.2** | **36.0** | **47.3** |

Teacher III has a very different head from Student III, while not as stand-alone accurate as Teacher IV, making it unable to provide enough guidance to Student III. These observations suggest that a smaller capacity gap between the student and the teacher may facilities knowledge transfer.

*The sequential order of the teachers plays a more critical role in the heterogeneous setting.* Table 12 (row 7-12) presents representative results with different orders or combinations of the teachers. Again, a proper progressive distillation (row 12) outperforms simultaneous distillation (row 7-9). Notably, it is necessary to start with Teacher I, since the capacity gap between Student III and Teacher I is minimal, with difference only on the depth of their ResNet backbones. These results confirm the importance of our curriculum-like progression to best benefit from multiple teachers.

**Training a student longer vs. distilling a student:** As another sanity check, Table 12 includes results of training Student III with more epochs without distillation (row 2-4). We can see that the first 12 additional epochs improve APs by 1%, but there are no significant improvements even if we train for a longer period. This shows the effectiveness of detector distillation.

**Distillation with different model resolutions:** In Table 12, we have performed distillation where the student and teacher models operated on the same input image resolution (*e.g.*, the standard resolution $1,333 \times 800$ on MS COCO). In practice, one way to further reduce the latency/runtime of the student is to operate on lower-resolution images. However, this poses additional challenges – with a teacher of high input resolution and a student of low input resolution, they become even more heterogeneous. Moreover, image resolution substantially affects object detection performance (Ashraf et al., 2016). Here, we are interested in performing distillation with models trained with images of different resolutions to further investigate the generalizability of our approach. More specifically, we use high-resolution models as teachers and low-resolution models as students, as shown in Table 13 (row 1-4).

In these experiments, the teacher and student feature maps have different *spatial resolution*. To tackle this, we simply upsample the spatial maps of the student and supervise the student with the teachers' features. Again, Table 13 shows that our approach is effective in this more challenging scenario. Our best performance is achieved by progressively distilling the student with its Teacher I-3, I-2, and I-1.

**Table 13:** Detectors trained with different input resolutions on the COCO dataset. We use a series of Teacher I variants: Teacher I-1 is trained with the standard input resolution of $1,333 \times 800$; Teacher I-2 is trained with $1,000 \times 600$ input; Teacher I-3 is trained with $666 \times 400$ input; and the student is trained with $333 \times 200$ input. We report the detection ('Box') and segmentation ('Mask') APs and runtime. We compare our distilled student with its teachers, and off-the-shelf ('OTS') student. Our approach is *effective with even more heterogeneous teacher and student models of different input resolutions*.

| ID | Model | Input Resolution | Box AP | $AP_{50}$ | $AP_{75}$ | $AP_S$ | $AP_M$ | $AP_L$ | Mask AP | $AP_{50}$ | $AP_{75}$ | $AP_S$ | $AP_M$ | $AP_L$ | Runtime (ms) |
|---|---|---|---|---|---|---|---|---|---|---|---|---|---|---|---|
| 1 | Teacher I-1 | $1333 \times 800$ | 38.2 | 58.8 | 41.4 | 21.9 | 40.9 | 49.5 | 34.7 | 55.7 | 37.2 | 18.3 | 37.4 | 47.2 | 31.5 |
| 2 | Teacher I-2 | $1000 \times 600$ | 37.2 | 57.7 | 40.5 | 19.1 | 40.9 | 50.4 | 33.6 | 54.3 | 35.9 | 15.6 | 37.0 | 47.7 | 24.9 |
| 3 | Teacher I-3 | $666 \times 400$ | 34.7 | 54.0 | 37.2 | 15.6 | 38.1 | 50.4 | 31.2 | 50.5 | 33.2 | 12.2 | 34.4 | 47.0 | 19.7 |
| 4 | Student (OTS) | $333 \times 200$ | 25.8 | 41.9 | 27.1 | 7.0 | 27.8 | 44.3 | 23.0 | 38.7 | 23.7 | 5.0 | 23.7 | 41.3 | 16.9 |
| 5 | Student (distilled) | $333 \times 200$ | **31.5** | **49.8** | **33.3** | **12.3** | **34.3** | **48.9** | **28.2** | **46.5** | **29.0** | **9.3** | **30.3** | **45.4** | 16.9 |

## D  GENERALIZABILITY TO OTHER DATASETS AND EVALUATION PROTOCOLS

In this section, we study the generalizability of our approach. As an extension from the gold-standard COCO benchmark, we evaluate our distilled student (trained on COCO) on another dataset, Argoverse-HD, and with another metric, streaming accuracy, and perform distillation on Argoverse-HD directly.

**Table 14:** Generalizability on Argoverse-HD. On the **left**, we report standard detection accuracy. 'OTS' and distilled students are trained on COCO. We observe 2% AP gains through distillation, even on novel testsets. On the **right**, we report streaming detection accuracy as defined in Li et al. (2020a), in the detection-only setting on a Tesla V100 GPU. The second column denotes the optimal input resolution (that maximizes streaming accuracy). First, we discover that a lighter model and full-resolution input is much more helpful than having an accurate but complex model that needs to downsize input resolution. Second, our proposed distillation approach further improves over the lightweight model.

| Model | | box AP | $AP_{50}$ | $AP_{75}$ | $AP_S$ | $AP_M$ | $AP_L$ |
|---|---|---|---|---|---|---|---|
| Stud. II | OTS | 32.7 | 52 | 34.5 | 14.7 | 35.8 | 52.8 |
| | Distilled | **34.4** | **54.2** | **35.9** | **15.0** | **36.8** | **57.7** |
| Stud. III | OTS | 28.9 | 48.8 | 30.0 | 12.8 | 31.3 | 49.2 |
| | Distilled | **30.6** | **49.7** | **31.8** | **12.9** | **32.6** | **51.9** |

| Detector | Input | AP | $AP_{50}$ | $AP_{75}$ | $AP_S$ | $AP_M$ | $AP_L$ |
|---|---|---|---|---|---|---|---|
| Cas. MRCNN50 (Li et al., 2020a) | 0.5× | 14.0 | 26.8 | 12.2 | 1.0 | 9.9 | 28.8 |
| MRCNN18 (Ours) | 1.0× | 23.7 | 44.8 | 22.6 | 10.4 | 23.1 | 37.8 |
| MRCNN18 (+ Distill) | 1.0× | **25.0** | **45.8** | **24.2** | **10.5** | **24.1** | **39.3** |

**Table 15:** Heterogenous distillation of Argoverse-HD detectors, where a student with ResNet-18 backbone is distilled with teachers with ResNet-50 backbones. We report the detection ('Box') APs and runtime. We compare our distilled student with its teachers, and off-the-shelf ('OTS') student. Our distilled student significantly improves the APs over the 'OTS' student by over 2%. Notably, our distilled student achieves detection accuracy that is *comparable with Teacher A but with only around third of the runtime*.

| ID | Model | Backbone | Neck | Method (Head) | Box AP | $AP_{50}$ | $AP_{75}$ | $AP_S$ | $AP_M$ | $AP_L$ | Runtime (ms) |
|---|---|---|---|---|---|---|---|---|---|---|---|
| 1 | Teacher A | ResNet-50 | FPN | Faster R-CNN | 29.6 | 48.2 | 30.5 | 16.4 | 33.1 | 45.1 | 79.2 |
| 2 | Teacher B | ResNet-50 | FPN | Cascade | 32.3 | 50.4 | 35.0 | 16.4 | 37.1 | 47.7 | 89.0 |
| 3 | Teacher C | ResNet-50 + SAC | RFP | Faster R-CNN | 32.9 | 51.0 | 35.5 | 17.6 | 33.7 | 52.9 | 230.8 |
| 4 | Teacher D | ResNet-50 + SAC | RFP | Cascade | 34.5 | 52.0 | 37.7 | 17.9 | 37.0 | 52.8 | 241.2 |
| 5 | Student (OTS) | ResNet-18 | FPN | Faster R-CNN | 27.1 | 48.1 | 27.5 | 14.4 | 31.2 | 40.0 | 29.3 |
| 6 | Student (distilled) | ResNet-18 | FPN | Faster R-CNN | **29.2** | **49** | **30.9** | **15** | **31.7** | **45.6** | 29.5 |

**Argoverse-HD** is a more challenging dataset than COCO due to higher resolution images and significantly more small objects. Constructed from the autonomous driving dataset Argoverse 1.1 (Chang et al., 2019), Argoverse-HD contains RGB video sequences and dense 2D bounding box annotations (1,260k boxes in total). It consists of 8 object categories, which are a subset of 80 COCO classes and are directly relevant to autonomous driving: person, bicycle, car, motorcycle, bus, truck, traffic light, and stop sign. There are 38k training images and 15k validation images. We report results on the validation images. We test the distilled models trained on COCO on Argoverse-HD *without re-training*. Table 14-left shows the generalizability of our approach.

**Streaming accuracy** is a recently proposed metric that simultaneously evaluates both the accuracy and latency of algorithms in an online real-time setting (Li et al., 2020a). The evaluator queries

the state of the world at all time instants, forcing algorithms to consider the amount of streaming data that must be ignored while processing the last frame. Following the setup proposed in Li et al. (2020a), we evaluate streaming AP in the context of real-time object detection for autonomous vehicles. Table 14-right shows our approach outperforming the prior results from Li et al. (2020a) by a dramatic margin. We find significant wins by using an exceedingly lightweight network (ResNet-18 based Mask R-CNN) that can process full-resolution images without sacrificing latency. Due to much higher quantities of small objects, high-reslution processing is more effective than deeper network structures. In addition, progressive distillation further improves performance.

**Direct distillation on Argoverse-HD:** After testing the distilled model which is trained on COCO, on the Argoverse-HD dataset (Li et al., 2020a) without re-training, we have shown the generalizability of the already-distilled models. Here we *directly distill* the student model on Argoverse-HD, using Faster R-CNN with a ResNet-18 backbone as the student model. As shown in Table 15, we use four teachers with ResNet-50 backbones (row 1-4), including Faster R-CNN (Ren et al., 2014) (Teacher A), Cascade R-CNN (Cai & Vasconcelos, 2018) (Teacher B), and DetectoRS (Qiao et al., 2021) (Teachers C & D).

The results are summarized in Table 15. Our best distillation performance is achieved when we first distill the student with a similar teacher (Teacher A), and then progressively distill with more powerful teachers (Teachers B, then C, and finally D). Overall, the bbox mAP is improved from 27.1% to 29.2%.

In addition, comparing with Table 14-left, the *absolute* performance of the teachers and students in Table 15 is lower. This is because here we use weaker teachers and student models (Faster R-CNN for fast experiments) than the models used in Table 14-left (Mask R-CNN). However, the *relative* improvement (between the distilled and OTS students) of box AP (2.1%) is larger than that in Table 14-left (1.7%), indicating that learning distillation directly on Argoverse-HD further improves the performance.

# E    MORE COMPARISON WITH PRIOR KNOWLEDGE DISTILLATION METHODS

The most profound difference between this work and most of the prior work on knowledge distillation is that prior work mainly focuses on the image classification task, while we address the object detection task. The detection task (and the associated model architectures) is much more complicated than the classification task. This makes the distillation methods developed in the context of classification often not directly applicable to detection. That is why dedicated distillation methods (Chen et al., 2017; Wang et al., 2019; Guo et al., 2021; Zhang & Ma, 2021; Dai et al., 2021; Guo et al., 2021; Yang et al., 2022a;b) need to be developed for the detection task in the literature. Here, we discuss the difference between our method and prior method on knowledge distillation in detail:

**Progressive distillation:** Mirzadeh et al. (2020) is related to our method, in the sense that this work progressively distills a student from multiple teachers (one teacher and several additional teacher assistants (TAs)). With one TA, the distillation process in Mirzadeh et al. (2020) contains three steps: 1) The TA is first distilled from the teacher; 2) The student is distilled from the TA; and 3) The student trained by the TA is further distilled from the teacher. When there are multiple TAs, the shallower TAs are distilled from deeper ones, so that they form a distillation path. However, our work is different from Mirzadeh et al. (2020) in three important ways:

- As mentioned above, Mirzadeh et al. (2020) focuses on image classification, while we study progressive distillation in the context of object detection. In our case, the transferred knowledge is no longer classification logits but intermediate features or structured predictions. This would require additional consideration and algorithmic designs to extend progressive distillation from image classification to object detection.
- Our strategy to construct the teacher sequence is a novel contribution and is fundamentally different from Mirzadeh et al. (2020). In our work, we propose a heuristic algorithm (Algorithm 1) based on the representation similarities between different models (Section 3.2), which automatically generates the teacher order. In Mirzadeh et al. (2020), a series of deep networks with increasing depths act as the student, the TA(s), and the teacher. One can intuitively determine the distillation sequence of TA(s) according to their increasing depths (which imply learning capacities). However in our case, there is a pool of teachers with diverse architectures and their relative

ordering is unknown. This challenge motivates us to design an algorithm to automatically decide the teacher order based on their representation similarities; importantly, the strategy in Mirzadeh et al. (2020) is not applicable in our task.

- Mirzadeh et al. (2020) is more cumbersome and time-consuming. An intermediate TA in Mirzadeh et al. (2020) needs to be first distilled from the teacher or a deeper TA, so the TAs have to be trained one by one. By contrast, all of our teachers (including the intermediate teachers and the final teacher) are trained independently. This makes the generation of our teachers parallelizable.

**Multi-teacher distillation:** The key difference lies in: These methods (You et al., 2017; Lan et al., 2018; Guo et al., 2020) use an ensemble of multiple teachers simultaneously to guide the student learning, while our work distills from multiple teachers sequentially, and we proposed a novel method to construct the appropriate teacher order. Empirically, we compare these two strategies, and demonstrate that our sequential progressive strategy outperforms the simultaneous strategy (via teacher ensemble by taking the average of teacher features) for object detection.

- This comparison is provided in Appendix B (Table 10) and Appendix C (Table 12). For example in Table 12, if we compare experiments with ID 7-9 (simultaneous distillation from teacher ensembles) vs. experiments with ID 10-12 (progressive distillation from teacher sequences), we find that progressive distillation is a better choice.
- Our performance superiority is because in the object detection task, the teacher's knowledge is transferred from intermediate features, rather than from final classification predictions. Thus, the ensemble of multiple teachers might provide conflicting supervision signals for the student, leading to performance interior to our progressive distillation.

**Online distillation, deep mutual learning:** Although the teacher model is also changing during online distillation (Yang et al., 2019a; Guo et al., 2020; Yao & Sun, 2020; Li et al., 2022), the principle of our sequential teachers is significantly different from online distillation for the following reasons:

- Strictly speaking, in online distillation, there is only one teacher – This teacher's architecture is fixed, and its weights keep updating in an online manner. By contrast, we have multiple teachers – These teachers have different architectures, and their weights are first trained independently, and then frozen in the progressive distillation process; in our progressive distillation, we switch the whole teacher model.
- The type of discrepancy between the student and the teacher is different for ours and online distillation. Online distillation often uses similar or even the same architecture for both the teacher and student models. Consequently, their capacities are at the same level, and they can evolve together. Our study is quite different: The key question we want to address is the capacity gap between the student and the teacher (the capacity gap is due to the architectural difference between the student and the teacher); and our solution is to progressively distill using other teachers with intermediate capacities.

**Other general distillation mechanisms:** These methods (Romero et al., 2015; Zagoruyko & Komodakis, 2017; Ahn et al., 2019) introduce other types of distillation mechanisms, but still consider the setting where only one single fixed teacher is involved. Different from these methods, we use multiple teachers to progressively transfer knowledge from them to the student. Meanwhile, we share some similarities with Romero et al. (2015); Zagoruyko & Komodakis (2017); Ahn et al. (2019) in that they are distilling knowledge from the activations of intermediate layers. Our simple feature-matching loss (Section 3.1) and other recent work in detector distillation (*e.g.*, CWD (Shu et al., 2021), FGD (Yang et al., 2022a), and MGD (Yang et al., 2022b)) are based on the "hint" distillation (learning from intermediate layers' outputs) from Romero et al. (2015).

## F  MORE RESULTS ON COMBINATION WITH STATE-OF-THE-ART DISTILLATION MECHANISMS

In this section, we include additional experimental results requested by the reviewers.

**Other object detectors:** In the main paper, we mainly use RetinaNet and Mask R-CNN as student detectors for a fair comparison with existing methods, because most of the prior work on detector distillation has primarily focused on RetinaNet and Mask R-CNN. Being commonly-used in real-

world applications, RetinaNet and Mask R-CNN represent the two important families of object detectors (single-stage and two-stage).

In principle, our work is general and applicable to different types of detection models, as our progressive distillation strategy is designed without assumption on particular types of detector architectures. To demonstrate this, we have performed an additional experiment on another detection model, RepPoints (Yang et al., 2019b). RepPoints is an anchor-free detector, with high efficiency-accuracy trade-off. Meanwhile, RepPoints has been experimented in state-of-the-art detector distillation methods such as MGD (Yang et al., 2022b), so we can provide an informative comparison. Therefore, we choose RepPoints as the detection model in this additional experiment. The student model is RepPoints/ResNet-50 (38.6 AP on COCO), and following state-of-the-art method MGD, we use RepPoints/ResNeXt-101 (44.2 AP) as the final teacher model. The intermediate teacher in our progressive distillation is RepPoints/ResNet-101 (40.5 AP). The following Table 16 shows the results obtained.

**Table 16:** Distillation of RepPoints (Yang et al., 2019b) detector. The student detector is RepPoints/ResNet-50, and the final teacher detector is RepPoints/ResNeXt-101. By using an intermediate teacher RepPoints/ResNet-101 for progressive distillation, we improve the student performance to **42.9%** AP.

| ID | Student | Distillation | Teacher(s) | AP | $AP_S$ | $AP_M$ | $AP_L$ |
|----|---------|--------------|-----------|-----|-----|-----|-----|
| 1 | RepPoints/ResNet-50 | None | None | 38.6 | 22.5 | 42.2 | 50.4 |
| 2 | | CWD (Shu et al., 2021) | RepPoints/ResNeXt-101 | 42.0 | 24.1 | 46.1 | 55.0 |
| 3 | RepPoints/ResNet-50 | FGD (Yang et al., 2022a) | RepPoints/ResNeXt-101 | 42.0 | 24.0 | 45.7 | 55.6 |
| 4 | | MGD (Yang et al., 2022b) | RepPoints/ResNeXt-101 | 42.3 | 24.4 | 46.2 | 55.9 |
| 5 | RepPoints/ResNet-50 | MGD (Yang et al., 2022b) + Progressive distillation (Ours) | RepPoints/ResNet-101 →RepPoints/ResNeXt-101 | **42.9** | **25.6** | **46.9** | **56.3** |

Our progressive distillation strategy improves the performance of the student detector by **4.3%** AP, which is also **0.6%** AP better than the previous state-of-the-art MGD. This result is achieved by only introducing an intermediate teacher and without increasing the training cost. This experiment further demonstrates that our progressive distillation strategy is general and can be applied to various detectors.

**Keep using the intermediate teacher:** In the main paper, we have discussed the capacity gap between the teacher and student and how to mitigate this gap via progressive distillation. One may question that, since the capacity gap between the *intermediate teacher* and the student is smaller, keeping using this intermediate teacher throughout the distillation procedure might also lead to a good student. Here we provide an exemplary experiment result (in addition to Figure 4-left) as an answer to this question: Only using the intermediate teacher is still suboptimal as compared with our proposed progressive distillation.

In this experiment, we use state-of-the-art distillation method MGD as the base method. We use RetinaNet/ResNet-50 (37.4 AP on COCO) as the student model, RetinaNet/ResNet-101 (38.9 AP) as the intermediate teacher model, and RetinaNet/ResNeXt-101 (40.8 AP) as the final teacher model. The capacity gap between the intermediate teacher and the student is smaller than that between the final teacher and the student. The following Table 17 shows the results.

**Table 17:** Distillation of RetinaNet detector. The capacity gap between the student (RetinaNet/ResNet-50) and the intermediate teacher (RetinaNet/ResNet-101) is smaller. Our progressive distillation is better than both distillation schemes that keep using the intermediate teacher (ID 3) or the final teacher (ID 2).

| ID | Student | Distillation | Teacher(s) | Training Schedule | AP |
|----|---------|--------------|-----------|-------------------|-----|
| 1 | RetinaNet/ResNet-50 | None | None | 2× | 37.4 |
| 2 | | | RetinaNet/ResNeXt-101 | 2× | 41.0 |
| 3 | RetinaNet/ResNet-50 | MGD (Yang et al., 2022b) | RetinaNet/ResNet-101 | 2× | 40.7 |
| 4 | | | RetinaNet/ResNet-101 | 1× | 40.2 |
| 5 | RetinaNet/ResNet-50 | MGD (Yang et al., 2022b) + Progressive distillation (Ours) | RetinaNet/ResNet-101 →RetinaNet/ResNeXt-101 | 1×+1× | **41.4** |

Keeping using the intermediate teacher (RetinaNet/ResNet-101) for a longer 2× training schedule indeed improves the performance from 40.2% AP to 40.7%, but it is still not better than directly

using the final teacher (RetinaNet/ResNeXt-101). Our progressive distillation, which first uses the intermediate teacher and then the final teacher for distillation, outperforms both direct distillation schemes and achieves **41.4%** AP performance. As always, we use the same total training time ($2\times$ training schedule) as direct distillation for a fair comparison. This experiment supports the conclusions that 1) employing an intermediate teacher throughout the distillation process is not a good option; and 2) our progressive distillation, which uses both the intermediate teacher and the best performing teacher sequentially, leads to the best student performance.

## G  IMPLEMENTATION DETAILS

We implement detectors and their distillation using the MMDetection codebase (Chen et al., 2019b). We train on 8 GPUs for 12 epochs for each distillation. For MS COCO, we use the standard input resolution of $1,333 \times 800$, with each GPU hosting 2 images. For Argoverse-HD, we use its much higher native resolution as the input at $1,920 \times 1,200$, with each GPU hosting 1 image. We use an initial learning rate of 0.01 (for RetinaNet students) or 0.02 (for Mask R-CNN students). We use stochastic gradient descent and a momentum of 0.9. For the simple feature-matching loss (see Section 3.1), we perform a grid search over the hyper-parameter $\lambda$. While the optimal values are dependent on the architectures of the teacher and student models, we find the performance is not very sensitive to $\lambda$ between 0.3 and 0.8. We set $\lambda = 0.5$ for RetinaNet students and $\lambda = 0.8$ for Mask R-CNN students.

When we combine our progressive distillation with state-of-the-art distillation mechanisms including CWD (Shu et al., 2021), FGD (Yang et al., 2022a), and MGD (Yang et al., 2022b) (Section 4.4), we strictly follow the publicly available implementation from their authors, and use an intermediate teacher (RetinaNet/ResNet-101 or Cascade Mask R-CNN/ResNet50-DCN) for progressive distillation. In the original implementation of FGD and MGD, an inheriting strategy (Kang et al., 2021) is utilized, which initializes the student with the teacher's neck and head parameters to train the student when they have the same head structure. In our progressive distillation, we adopt this inheriting strategy only once for the first teacher.

For the Transformer-based teachers, we use Swin Transformer backbone, which has a hierarchical architecture and shares the "same feature map resolutions as those of typical convolutional networks (*e.g.*, ResNet-50)" (Liu et al., 2021). Following the original implementation of the Swin Transformer, the backbone is equipped with an FPN neck, so the number of neck feature channels is the same as the student. As a result, Swin Transformer based teachers can be used like some other convolution-based teachers without the feature map matching function ($r(\cdot)$ in Section 3.1).

