# OpenReview forum: "Learning Lightweight Object Detectors via Progressive Knowledge Distillation"
_ICLR.cc/2023/Conference — Submitted to ICLR 2023_

### Official Review · Reviewer_zfUg · 2022-10-22

**Confidence:** 4
**Correctness:** 3
**Technical Novelty And Significance:** 3
**Empirical Novelty And Significance:** 3
**Recommendation:** 8

**Clarity, Quality, Novelty And Reproducibility:**

Clarity and Quality:
In general, the clarity and quality of this paper are good.

Novelty:
To the best knowledge, the paper provides some novelty ideas for this area.


**Strength And Weaknesses:**

Strengths:
1. The sequential strategy is inspiring and of interest for knowledge distillation for object detection and segmentation.
2. The performance of the proposed method is great, as shown in the results.
3. Comparison between the sequential way and simultaneous way is of interest.

Weakness:
1. Compared to other studies in the paper, the introduction to the distillation from the transformer-based model to the convolutional-based model is simple. For example, it seems that the transformer-based model is not combined with other models as teacher models. It would be better to provide a possible way or give some discussions in the paper.
2. How to address a large k, i.e., the maximum number of selected teachers, or how to pick a good k is not given or well-discussed. In practice, it may happen when users have a large pool of teachers.

**Summary Of The Paper:**

This paper proposes a progressive distillation strategy for object detection and instance segmentation models, i.e., multiple teacher models transfer knowledge to a student model in a sequential manner. A heuristic method is also given to choose the order of teacher models and its efficacy is demonstrated by experiments. A method for distilling knowledge from transformer-based teachers to convolutional-based students is simply introduced. Many experiments are done to verify the capability of the proposed strategy and ablation studies are provided.

**Summary Of The Review:**

The reviewer generally thinks the paper is written in sound quality as specified by the strengths above. It proposes a sequential manner for knowledge distillation. However, it would be recommended that authors address the weakness mentioned above and also the following comments:

1. In table 7, for length 4, there is no bold sequence marked. Is this a result of algorithm 1? Please add some explanations.
2. Maybe give a concrete example of $r$ in (1) for two concrete models in a heterogeneous case. It also would be great to see how the transformer-based model is addressed here.

-------------------
Update: I believe the authors well-addressed my comments.

---

> ### Author Response · Authors · 2022-11-08
> **Response to Reviewer zfUg**
>
> We appreciate the reviewer for the insightful comments and suggestions. Here we address all the concerns.
>
> - Transformer-based teacher:
>
>     In fact, the Transformer-based teacher used in our work is not treated differently from other teachers. We apologize for the confusion, and we will include the details regarding this Transformer-based distillation in the revision.
>
>     Specifically, in this work, we used teachers based on the Swin Transformer, which has a hierarchical architecture and shares the “same feature map resolutions as those of typical convolutional networks (e.g.. ResNet-50)” [Ref1]. Following the original implementation of the Swin Transformer, the backbone is equipped with an FPN neck, so the number of neck feature channels is the same as the student. As a result, Swin Transformer based teachers can be used like some other convolution-based teachers without the feature map matching function ($r(\cdot)$ in Equation 1). In our progressive distillation, we find distillation from first Swin-Tiny and then Swin-Small yields a better performance for a RetinaNet/ResNet-50 student (Table 5).
>
> - Deciding $k$ & no bold sequence marked in Table 7 (length 4):
>
>     Designing an algorithm that can determine the optimal $k$ is an interesting future direction. Indeed in our current Algorithm 1, we leave the **maximum** number of selected teachers as an input chosen by the user. In practice, one can decide this $k$ according to the training budget: For each teacher, the training should be sufficiently long (say, “1x” training schedule) for knowledge transfer. Then the maximum number of teachers should be $k$, if the total training budget is $k$ x.
>
>     Meanwhile, our Algorithm 1 may terminate early and return a sequence shorter than $k$, if the adaptation cost from the student to the first teacher is already very small – see Line 4-5 in Algorithm 1; this also explains why there is not a bold sequence in Table 7, k=4. If one does not care about the training budget and just wants to check the optimal teacher sequence, then $k=N$ can be set and a teacher sequence (probably not using all teachers in the pool) with minimal adaptation costs along the trajectory will be returned.
>
> - Example of $r(\cdot)$ in heterogeneous distillation:
>
>     Here we provide two examples of $r(\cdot)$ in heterogeneous distillation from Table 4:
>     - For experiments with ID 1 & 2, the teachers have ResNet-50 backbones, whose number of feature channels is 4 times as much as ResNet-18 in the students. The mapping $r(\cdot)$ is implemented as $1\times 1$ convolutional layers at each stage.
>     - For experiments with ID 3 & 4, the teachers have larger input resolutions than the student. The mapping $r(\cdot)$ is implemented as an upsampling layer.
>
>     As mentioned above, we do not need to transform the feature maps for Swin Transformer based teachers, since they share the same dimensionality with the students’ feature maps. $r(\cdot)$ is simply an identity function.
>
> Thank you again for your valuable review! We are looking forward to your further feedback.
>
> [Ref1] Ze Liu, Yutong Lin, Yue Cao, Han Hu, Yixuan Wei, Zheng Zhang, Stephen Lin, and Baining Guo. Swin transformer: Hierarchical vision transformer using shifted windows. In ICCV, 2021.

---

### Official Review · Reviewer_R4rk · 2022-10-25

**Confidence:** 4
**Correctness:** 3
**Technical Novelty And Significance:** 3
**Empirical Novelty And Significance:** 3
**Recommendation:** 6

**Clarity, Quality, Novelty And Reproducibility:**

Clarity, Quality: Good. The paper is well-written and easy to follow.

Novelty: Good. The proposed method improves the KD performance in object detection by using the capacity of different teacher models sequentially.

Reproducibility: Good. The supplementary code is provided.

**Strength And Weaknesses:**

**Strengths**
+ Novel idea. The authors propose a heuristic algorithm to produce a near-optimal sequence of multiple teachers, even not the best, still outperforms multiple KD approaches distilled from a single teacher.
+ Promising results and convincing ablation studies with multiple settings for object detection.

**Weaknesses**

However, there are still some concerns to be addressed:

- Even though the student can get better results with the selected order of the teachers by Algorithm 1, sometimes it is not the best, especially with more teachers. Could the author please provide some potential explanation on this? In other world,  how reliable the proposed Algorithm 1 is?

- As stated in Section 4.1, the student is initialized by an off-the-shell student for the proposed KD method. Does the student use the same strategy of initialization for exp 3 and 7 (Directly distilled by Teacher IV) and blue bars in Figure 4? Moreover, as for Figure 4, it is not quite fair to compare the performance of 2x training with 2 1x training. Please clarify it.

- The capacity gap between the student and the best performance teacher may a problem for knowledge distillation. Maybe the intermediate teacher is better to be distilled for the student. Thus, it will be interesting to compare with the distilled results from the intermediate teachers in Tables 3, 4 and 5, and Figure 4.

**Minor comments**

It is worth discussing a related work [A], which also makes use of several teachers. It is interesting to the research line of KD in object detection.

**Reference**

[A] Guo. et al. Distilling Image Classifiers in Object Detectors. NeurIPS2021


**Summary Of The Paper:**

The paper proposes the progressive knowledge distillation method for object detection by distilling knowledge from multiple teachers. The main contribution of this work is designing a heuristic algorithm based on the correlation of feature representations to generate the proper sequence of several teachers given a student model. Comprehensive experiments and ablation studies on the COCO dataset evidence the effectiveness of the proposed method.

**Summary Of The Review:**

The paper is in good shape with extensive experiments, which show promising results. However, there are still some concerns. Please clarify them.

---

> ### Author Response · Authors · 2022-11-08
> **Response to Reviewer R4rk**
>
> We appreciate the reviewer for the insightful comments and suggestions. Here we address all the concerns.
>
> - Algorithm 1 not optimal:
>
>     We agree with the reviewer that our heuristic Algorithm 1 cannot guarantee the optimality in theory. Especially, as the pool of candidate teachers grows, the number of possible teacher sequences increases exponentially. Due to the variance and instability in student training, it is less likely to precisely produce the best sequence when the number of teachers is larger. However, our Algorithm 1 is still reliable and produces a teacher sequence that leads to strong performance – this is demonstrated from two aspects:
>
>     - Outperforming alternative strategies: As shown in Appendix A, we have compared our Algorithm 1 with other possible algorithms (e.g., shortest path), and Algorithm 1 is still the best choice.
>
>     - Close to the optimal performance: When we **compare the absolute performance values, even if our produced teacher sequence is not optimal, it is very close to the upper bound performance**. For example in Table 2, when k is larger, our teacher sequence ranks 7th, but the resulting student performance (37.9 AP) is very close to the best possible performance (38.2 AP on COCO).
>
> - Student initialization:
>
>     Our comparison is fair.
>
>     - In Table 3 where we compare simple feature-matching distillation with and without our progressive distillation, both settings (ID 3, 4, 7, 8) initialize the student using off-the-shelf weights.
>
>     - In Figure 4 where we compare SOTA distillation mechanisms with and without our progressive distillation, we follow their original implementation and initialize students in the same way. For example for MGD, the student’s ResNet backbone is loaded from standard ImageNet pre-training, and the detector head is inherited from the teacher’s weights. This is the same for both direct distillation and our progressive distillation. We apologize for the confusion, and we will include the implementation details of the student initialization in the revision.
>
>     - “2x training” and “two 1x training” take roughly the same amount of time (in fact, our “two 1x training” is slightly faster because the first teacher takes less computation than the final teacher), so they are comparable. We have to apply this “two 1x training” schedule because we have different teachers as learning targets. Moreover, our previous experiments find “two 1x training” is not better than “2x training” when a fixed teacher is used (direct distillation). For example, when distilling the RetinaNet student only with the DetectoRS teacher, “2x training” is better than “two 1x training” (39.5 AP vs. 39.3 AP on COCO).
>
> - Keep using intermediate teachers:
>
>     Thanks for suggesting this interesting experiment. We will provide additional results regarding using a mid-capacity teacher for direct distillation, by the end of this discussion period. We will also include the results in the final revision.
>
> - Related work [A] Distilling Image Classifiers in Object Detectors:
>
>     Thanks for pointing out this related work [A] and we will cite it in the revision. In [A], an image classifier (in contrast to an object detector in other methods) is used as the teacher model, and it provides both classification and localization knowledge to the student detector. However, we do not find the usage of multiple teachers when training one student in [A]. Would the reviewer mind elaborating on this comment?
>
> Thank you again for your valuable review! We are looking forward to your further feedback.

---

> > ### Author Response · Authors · 2022-11-18
> > **Update: Additional Experiments on Keeping Using Intermediate Teacher in Distillation**
> >
> > Here we provide an update regarding the experiment previously requested by the reviewer. In our initial submission, we discussed that “the capacity gap between the student and the best performing teacher may be a problem for knowledge distillation,” as mentioned by the reviewer. The reviewer suggested that keeping using the intermediate teacher might be a better choice for distilling the student detector.
> >
> > Due to limited time and computation resources, we have conducted one experiment as an addition to Figure 4-left in our initial submission, to show that only using the intermediate teacher is still suboptimal as compared with our proposed progressive distillation.
> >
> > In this experiment, we use state-of-the-art distillation method MGD as the base method. We use RetinaNet/ResNet-50 (37.4 AP on COCO) as the student model, RetinaNet/ResNet-101 (38.9 AP) as the intermediate teacher model, and RetinaNet/ResNeXt-101 (40.8 AP) as the final teacher model. The capacity gap between the intermediate teacher and the student is smaller than that between the final teacher and the student.
> > | Model               | Method                                | Teacher(s)                                    | Training Schedule | AP   |
> > |---------------------|---------------------------------------|-----------------------------------------------|-------------------|------|
> > | RetinaNet/ResNet-50 | No distillation                       | None                                          | 2x                | 37.4 |
> > | RetinaNet/ResNet-50 | MGD + Direct distillation             | RetinaNet/ResNeXt-101                         | 2x                | 41.0 |
> > | RetinaNet/ResNet-50 | MGD + Direct distillation             | RetinaNet/ResNet-101                          | 2x                | 40.7 |
> > | RetinaNet/ResNet-50 | MGD + Direct distillation             | RetinaNet/ResNet-101                          | 1x                | 40.2 |
> > | RetinaNet/ResNet-50 | MGD + Progressive distillation (Ours) | RetinaNet/ResNet-101 -> RetinaNet/ResNeXt-101 | 1x + 1x           | **41.4** |
> >
> > Keeping using the intermediate teacher (RetinaNet/ResNet-101) for a longer 2x training schedule indeed improves the performance from 40.2% AP to 40.7%, but it is still not better than directly using the final teacher (RetinaNet/ResNeXt-101). Our progressive distillation, which first uses the intermediate teacher and then the final teacher for distillation, outperforms both direct distillation schemes and achieves **41.4%** AP performance. As always, we use the same total training time (2x training schedule) as direct distillation for a fair comparison.
> >
> > In addition to this experiment, we are also continuing with more experiments and will include all the results of direct distillation using the intermediate teacher in Tables 3, 4 and 5 in future revision.
> >
> > We believe that our experiment supports the conclusions that 1) employing an intermediate teacher throughout the distillation process is not a good option; and 2) our progressive distillation, which uses both the intermediate teacher and the best performing teacher sequentially, leads to the best student performance.

---

> > > ### Comment · Reviewer_R4rk · 2022-11-20
> > > **Thank you for the response.**
> > >
> > > Thank you for the response and more experiements.
> > >
> > > I still have some concerns about the two 1x and 2x. Basically, they are different from the schedules of learning rate, warm_up, etc. It is hard to say they are comparable without the numbers. If "2x training” is better than “two 1x training”, why did the authors conduct their experiments with two 1x, not using 2x training? I think it will be more convincing compare two 1x with two 1x, or 2x with 2x.

---

> > > > ### Author Response · Authors · 2022-11-23
> > > > **Update: Additional Clarification about Different Training Schedules**
> > > >
> > > > Thank you for reading our response. Here, we provide a more detailed clarification about the different training schedules to address the reviewer’s concern.
> > > >
> > > > - Indeed, “two 1x” and “2x” are different training schedules, reflected in how they change the learning rate during the whole distillation process. For the “two 1x” schedule, the learning rate is restarted halfway, while “2x” does not restart it.
> > > >
> > > > - We empirically find: On the one hand, “two 1x” is marginally better than “2x” for our progressive distillation, where we need to switch the teacher halfway and restart the learning rate accordingly. On the other hand, “2x” is better for direct distillation baselines, since the training target is fixed.
> > > >
> > > > - To observe this phenomenon, we have conducted more experiments in the setting of distilling a RetinaNet/ResNet-50 student with MGD, as shown below.
> > > >
> > > >     | Student             | Method                                | Teacher(s)                                    | Schedule | AP   |
> > > >     |---------------------|---------------------------------------|-----------------------------------------------|----------|------|
> > > >     | RetinaNet/ResNet-50 | MGD + Direct distillation             | RetinaNet/ResNeXt-101                         | 2x       | 41.0 |
> > > >     | RetinaNet/ResNet-50 | MGD + Direct distillation             | RetinaNet/ResNeXt-101                         | 1x + 1x  | 40.8 |
> > > >     | RetinaNet/ResNet-50 | MGD + Progressive distillation (Ours) | RetinaNet/ResNet-101 -> RetinaNet/ResNeXt-101 | 2x       | 41.3 |
> > > >     | RetinaNet/ResNet-50 | MGD + Progressive distillation (Ours) | RetinaNet/ResNet-101 -> RetinaNet/ResNeXt-101 | 1x + 1x  | 41.4 |
> > > >
> > > >     While “two 1x” and “2x” bring some performance change, they do not change the conclusion that our progressive distillation is a better strategy. In fact, if we did the comparison with a fixed “two 1x” training schedule, the relative improvement would be larger (0.6% AP).
> > > >
> > > > - As we have mentioned in the previous response, **“two 1x” and “2x” schedules require almost the same amount of total training time, so we believe they are comparable. We treat the training schedule as a tunable hyper-parameter and choose the best for either the baseline or our method. Therefore, our comparison is indeed fair**.

---

### Official Review · Reviewer_1tUC · 2022-10-28

**Confidence:** 5
**Correctness:** 4
**Technical Novelty And Significance:** 3
**Empirical Novelty And Significance:** 3
**Recommendation:** 5

**Clarity, Quality, Novelty And Reproducibility:**

Clarity: good.
Quality: The organization and architecture of the paper is not very clear, especially the experiment part.
Novelty: somewhat, but a little incremental. The expression can be improved.
Reproducibility: I think it is easy to follow. But the code in appendix is not complete and not conducted by the common MMDetection framework like most works, e.g. FGD, MGD.


**Strength And Weaknesses:**


Strength:
1. The idea of progressively transferring knowledge from a sequence of teachers to a lightweight detector is somewhat novel.
2. It represents the first effort to distill knowledge from Transformer-based teacher detectors to convolution-based students.
3. It shows the performance gain comes from better generalization rather than better optimization.

Weaknesses:
1. Why not compare with other general[4,5,6] / multi-teacher[9,10] / progressive[1-3,7,8] KD methods?
2. Please clarify the difference between the proposed method and online KD[] algorithms. And it's better to give more experiments and analysis to highlight the strength of your idea. I think that the principle of sequence teacher is similar to online distillation.
3. The ranking of teachers seems to require empirical design. It's better to give more theoretical analysis.
4. Another serious concern is that the pre-training of multiple teachers can be cumbersome and time-consuming, making the pipeline complicated and unpractical.
5. Since the author propose to transfer knowledge between Transformer and CNN, why not adopt Transformer-based student detectors?
6. There is lack of some important baselines [11-16].

## References

[1] Improved Knowledge Distillation via Teacher Assistant: Bridging the Gap Between Student and Teacher. Mirzadeh et al. arXiv:1902.03393

[2] Snapshot Distillation: Teacher-Student Optimization in One Generation. Yang, Chenglin et al. CVPR 2019

[3] Online Knowledge Distillation by Temporal-Spatial Boosting. Li, Chengcheng et al. WACV 2022

[4] Fitnets: Hints for thin deep nets. Romero, Adriana et al. arXiv:1412.6550

[5] Paying more attention to attention: Improving the performance of convolutional neural networks via attention transfer. Zagoruyko et al. ICLR 2017

[6] Variational Information Distillation for Knowledge Transfer. Ahn, Sungsoo et al. CVPR 2019

[7] Online Knowledge Distillation via Collaborative Learning. Guo, Qiushan et al. CVPR 2020

[8] Knowledge Transfer via Dense Cross-layer Mutual-distillation. ECCV 2020

[9] Learning from Multiple Teacher Networks. You, Shan et al. KDD 2017

[10] Knowledge distillation by on-the-fly native ensemble. Lan, Xu et al. NeurIPS 2018

[11] Learning efficient object detection models with knowledge distillation. Chen, Guobin et al. NeurIPS 2017

[12] Distilling Object Detectors with Fine-grained Feature Imitation. Wang, Tao et al. CVPR 2019

[13] Enabling Incremental Knowledge Transfer for Object Detection at the Edge. arXiv:2004.05746

[14] General Instance Distillation for Object Detection. Dai, Xing et al. CVPR 2021

[15] Distilling Image Classifiers in Object Detectors. Guo, Shuxuan et al. NeurIPS 2021

[16] Improve Object Detection with Feature-based Knowledge Distillation: Towards Accurate and Efficient Detectors. ICLR 2021


**Summary Of The Paper:**

The paper proposes a simple progressive knowledge distillation framework with a sequence of teachers for detectors.

**Summary Of The Review:**

The work is somewhat simple yet effective, my major concern is the novelty and experiments.
If the authors can make the experimental part simple and clear, and highlight the competitiveness of the results with more SOTAs, I will consider improve the score.

---

> ### Author Response · Authors · 2022-11-08
> **Response to Reviewer 1tUC (part 1)**
>
> We appreciate the reviewer for the insightful comments and suggestions. Here we address all the concerns.
>
> - Difference with prior KD work:
>
>     First, we sincerely appreciate the reviewer for sharing this extensive list of related work. We will cite all of them and include detailed discussion in the revision.
>
>     Second, we would like to clarify that most of the papers ([1-10]) are on knowledge distillation for the image **classification** task, while we address the object **detection** task. The detection task (and the associated model architectures) is much more complicated than the classification task. This makes the distillation methods developed in the context of classification often not directly applicable to detection. That is why dedicated distillation methods need to be developed for the detection task in the literature. We would greatly appreciate the reviewer to take this important distinction when comparing our work with prior work.
>
>     Below we discuss the difference between our method and the suggested papers in detail:
>
>     - Progressive distillation [1]: [1] is related to our method, in the sense that [1] progressively distills a student from multiple teachers (one teacher and several additional teacher assistants (TAs)). With one TA, the distillation process in [1] contains three steps: 1) The TA is first distilled from the teacher; 2) The student is distilled from the TA; and 3) The student trained by the TA is further distilled from the teacher. When there are multiple TAs, the shallower TAs are distilled from deeper ones, so that they form a distillation path. However, our work is different from [1] in three important ways:
>         - As mentioned above, [1] focuses on image classification, while we study progressive distillation in the context of object detection. In our case, the transferred knowledge is no longer classification logits but intermediate features or structured predictions. This would require additional consideration and algorithmic designs to extend progressive distillation from image classification to object detection.
>         - Our strategy to construct the teacher sequence is a novel contribution and is fundamentally different from [1]. As mentioned by **Reviewer XUDr**, our paper “proposes to choose the policy of student-teacher in knowledge distillation by using the cost of linear regression on the validation set, which is novel in this domain.” Also, as mentioned by **Reviewer zfUg**, “the sequential strategy is inspiring and of interest for knowledge distillation for object detection and segmentation.” More specifically, in [1], a series of deep networks with increasing depths act as the student, the TA(s), and the teacher. One can intuitively determine the distillation sequence of TA(s) according to their increasing depths (which imply learning capacities). However in our case, there is **a pool of teachers with diverse architectures and their relative ordering is unknown**. This challenge motivates us to design an algorithm to automatically decide the teacher order based on their representation similarities; importantly, the strategy in [1] is not applicable in our task.
>         - [1] is more cumbersome and time-consuming. An intermediate TA in [1] needs to be first distilled from the teacher or a deeper TA, so the TAs have to be trained one by one. By contrast, all of our teachers (including the intermediate teachers and the final teacher) are trained independently. This makes the generation of our teachers parallelizable.
>
>     - Multi-teacher distillation [7, 9, 10]: The key difference lies in: These methods [7, 9, 10] use an ensemble of multiple teachers **simultaneously** to guide the student learning, while our work distills from multiple teachers **sequentially**, and we proposed a novel method to construct the appropriate teacher order. Empirically, in the original submission we compared these two strategies, and demonstrated that our sequential progressive strategy outperforms the simultaneous strategy (via teacher ensemble by taking the average of teacher features) for object detection.
>         - This comparison was provided in Appendix B (Table 10) and Appendix C (Table 12), due to space limit. For example in Table 12, if we compare experiments with ID 7-9 (simultaneous distillation from teacher ensembles) vs. experiments with ID 10-12 (progressive distillation from teacher sequences), we find that progressive distillation is a better choice.
>        - Our performance superiority is because in the object detection task, the teacher's knowledge is transferred from intermediate features, rather than from final classification predictions. Thus, the ensemble of multiple teachers might provide conflicting supervision signals for the student, leading to performance interior to our progressive distillation.

---

> > ### Author Response · Authors · 2022-11-08
> > **Response to Reviewer 1tUC (part 2)**
> >
> > - Difference with prior KD work (continued):
> >
> >     - Online distillation, deep mutual learning [2, 3, 7, 8]: Although the teacher model is also changing during online distillation [2, 3, 7, 8], the principle of our sequential teachers is significantly different from online distillation for the following reasons:
> >         - Strictly speaking, in online distillation, there is **only one teacher** – This teacher’s architecture is **fixed**, and its weights keep updating in an online manner. By contrast, we have **multiple teachers** – These teachers have different architectures, and their weights are first trained independently, and then frozen in the progressive distillation process; in our progressive distillation, we switch the whole teacher model.
> >         - The type of discrepancy between the student and the teacher is different for ours and online distillation. Online distillation often uses similar or even the same architecture for both the teacher and student models. Consequently, their capacities are at the same level, and they can evolve together. Our study is quite different: The key question we want to address is the **capacity gap** between the student and the teacher (the capacity gap is due to the architectural difference between the student and the teacher); and our solution is to progressively distill using other teachers with intermediate capacities.
> >
> >     - Other general distillation mechanisms [4, 5, 6]: These methods [4, 5, 6] introduce other types of distillation mechanisms, but still consider the setting where **only one single fixed teacher** is involved. Different from these methods, we use multiple teachers to progressively transfer knowledge from them to the student. Meanwhile, we share some similarities with [4, 5, 6] in that they are distilling knowledge from the activations of intermediate layers. Our simple feature-matching loss (Section 3.1) and other recent work in detector distillation [11-16] are based on the “hint” distillation (learning from intermediate layers’ outputs) from [4].
> >
> > - Theoretical analysis:
> >
> >     We thank the reviewer for the suggestion. This is an empirical rather than theoretical work. Establishing satisfactory theoretical analysis is undoubtedly important but hard. This is particularly difficult for our task of object detection where complex detection models are involved. On the other hand, we believe that our empirical analysis is important and has demonstrated the effectiveness of our strategy for ranking teachers. We would like to kindly note that the distillation strategies for object detection proposed in most of the papers suggested by the reviewer are also evaluated mainly empirically without theoretical analysis. Further theoretical analysis is an excellent direction for future research.
> >
> > - Pre-training of teachers:
> >
> >     We agree with the reviewer that distillation from multiple teachers introduces additional computation overhead, compared with distillation from a single teacher. However, compared with the related work suggested by the reviewer that also utilizes multiple teachers, our overhead on teacher pre-training is either comparable (e.g., to multi-teacher distillation [7, 9, 10]) or significantly lower (e.g., than progressive distillation [1]). In particular, as mentioned above, **our teachers can be trained independently in parallel** – such independence simplifies the pre-training of our teachers, and such parallelizability speeds up our approach in practice. With respect to this computation aspect, our method is thus significantly better than [1]: [1] needs distillation between teachers, making their teachers have to be trained one by one.
> >
> >     In addition, we would like to emphasize that for knowledge distillation in practice, the end goal is to learn a high-accuracy and lightweight student that can be deployed in resource-constrained scenarios (e.g., self-driving cars, robotic perception systems). Often, the computation resource for **training** the student is not a major concern, where heavy teachers and even ensemble of multiple teachers are typically exploited in the literature to boost the performance. Moreover, in some cases, one may directly use some already-trained models as the teachers. For example, in our implementation, we did not need to train the teacher models on our own, but simply loaded the weights provided by publicly available libraries like MMDetection.

---

> > > ### Author Response · Authors · 2022-11-13
> > > **Response to Reviewer 1tUC (part 3)**
> > >
> > > - Other baselines [11-16]:
> > >
> > >     We thank the reviewer for the suggestion. While we did not compare with [11-16], in the original submission we compared with and outperformed baselines that are stronger than [11-16].
> > >
> > >     Note that our progressive distillation is orthogonal to previous efforts in detection distillation that design various distillation losses including [11-16], and it is a general strategy that can be readily combined with them. Specifically, in Section 4.4, we combined our progressive distillation strategy with three state-of-the-art object detection distillation methods and showed even better results: CWD (ICCV 2021), FGD (CVPR 2022), and MGD (ECCV 2022). These most recent methods have already outperformed prior work [11-16], so we omitted the comparison with [11-16] due to space limit.
> > >
> > >     In addition, per the reviewer’s request, the table below shows an experiment on combining ours with [16] ([15] and [16] are comparable, and both [15] and [16] outperform [11-14]). Our progressive distillation improves the student performance for [16]:
> > >
> > >     | Model                | Method                          | AP       | AP$_{50}$ | AP$_{75}$ | AP$_S$ | AP$_M$ | AP$_L$ |
> > >     |----------------------|---------------------------------|----------|-----------|-----------|--------|--------|--------|
> > >     | RetinaNet/ResNet-50  | [16] + Direct distillation      | 39.6     | 58.8      | 42.1      | 22.7   | 43.3   | 52.5   |
> > >     | RetinaNet/ResNet-50  | [16] + Progressive distillation | **40.2** | 59.4      | 43.1      | 21.7   | 43.5   | 55.6   |
> > >     | Mask R-CNN/ResNet-50 | [16] + Direct distillation      | 41.3     | 61.9      | 45.1      | 23.3   | 44.8   | 55.3   |
> > >     | Mask R-CNN/ResNet-50 | [16] + Progressive distillation | **41.6** | 62.0      | 45.4      | 23.3   | 44.7   | 55.9   |
> > >
> > > - Transformer-based students:
> > >
> > >     We did not include Transformer-based students in the original submission for the following three reasons:
> > >
> > >     - Consistent with practical scenarios: The end goal of this work and knowledge distillation in general is to develop high-accuracy, lightweight object detectors. To this end, the student detectors are typically efficient detectors like RetinaNet with a ResNet backbone. Although there has been some recent research focusing on designing computationally efficient architectures for Transformers, the widely adopted vision Transformers (e.g., ViT and Swin Transformer) are still slower than their convolution-based counterparts (e.g., ResNet and EfficientNet), because convolution operations are highly optimized on GPUs. Therefore, ResNet-based students are more favored in developing lightweight object detectors and are thus used in our evaluation.
> > >
> > >     - The key question we want to address is the **capacity gap** between the student and the teacher. If one has to distill knowledge from a high-accuracy Transformer-based teacher (for better performance) to a lightweight convolution-based student (for better efficiency), then the capacity gap is unavoidable and must be mitigated. In such a scenario, our progressive distillation is a helpful solution. In fact, we showed progressive distillation is critical to the success of knowledge transfer from a Swin Transformer teacher to a ResNet student in Table 5.
> > >
> > >     - Compared with distillation from homogeneous architectures (e.g., between Transformers as suggested by the reviewer), distillation from heterogeneous architectures (e.g., between Transformer and CNN as what we evaluated) is more challenging, which demonstrates the effectiveness of our approach.
> > >
> > >    In addition, per the reviewer’s request, we are currently experimenting on distillation between a Transformer-based student and Transformer-based teachers, to show that our approach is still effective in this setting. We will include the results in the final revision. If time allows, we will provide them by the end of this discussion period.

---

> > > > ### Author Response · Authors · 2022-11-13
> > > > **Response to Reviewer 1tUC (part 4)**
> > > >
> > > > - Organization of experiments:
> > > >
> > > >     We thank the reviewer for the suggestion. Below we clarify how we organized the experiment part in our original submission, and we hope this will clarify the reviewer’s confusion. We would greatly appreciate the reviewer’s suggestion on the specific organization, and we are very happy to modify the organization accordingly in the revision.
> > > >
> > > >     The focus of our experiments is to validate the effectiveness of our progressive distillation strategy from complementary perspectives. Therefore, the experiment part was organized for this purpose.
> > > >
> > > >     - In Section 4.1, we construct a resource-controlled setting where we have a pool of teachers and can enumerate *all possible* teacher orders, to verify our proposed heuristic algorithm can consistently generate near-optimal teacher orders.
> > > >
> > > >     - In Sections 4.2 and 4.3, based on the verified Algorithm 1, we distill various students with sufficiently long training schedules and show the benefit of progressive distillation.
> > > >
> > > >     - In Section 4.4, we further combine our progressive distillation strategy with state-of-the-art object detection distillation mechanisms (which are orthogonal to our contribution), and provide improved results and show the generalizability of our progressive distillation strategy (which is agnostic to these detection distillation mechanisms).
> > > >
> > > >     - In Section 4.5, we use a loss landscape analysis to understand the generalization gain of progressive distillation.
> > > >
> > > >     - To summarize, we start with the verification of our near-optimal teacher orders, then apply the progressive distillation strategy in multiple settings, combine it with state-of-the-art methods for optimal performance, and finally provide an analysis for deeper understanding. We believe that every part is important to support our progressive distillation.
> > > >
> > > > - Code:
> > > >
> > > >     Our code provided in the supplementary material is indeed based on MMDetection. We believe that our code is complete – In our supplementary material, we included additional files that define distillation models, configurations, and training routines which can be used without modifying MMDetection. One can directly use the code within the MMDetection codebase to reproduce our results. Please also refer to the README file for more information.
> > > >
> > > > Thank you again for your valuable review! We are looking forward to your further feedback.

---

> > ### Comment · Reviewer_1tUC · 2022-11-20
> > **Difference with prior KD work**
> >
> > Thanks for your response. But I still claim that though the works[1-10] are proposed for classification, they are general KD methods can can be applied to more tasks. For example, the vanilla KD proposed by Hinton has been extended to various tasks. Especially, there is not difference if using the feature distillation methods[4-6] on the backbone/encoder.  These classic methods has been proven simple yet efficient. So I think it's necessary to compare with them.

---

> > > ### Author Response · Authors · 2022-11-25
> > > **Update: Additional Experiment on Prior General KD Method**
> > >
> > > We thank the reviewer for the follow-up question. To address the reviewer’s further concern, we performed another experiment based on a prior knowledge distillation method. We choose Variational Information Distillation (VID) [6], because it is the latest and has proven the most excellent image classification performance among the previous general distillation methods [4-6] listed by the reviewer.
> > >
> > > We use RetinaNet/ResNet-50 as the student and RetinaNet/ResNeXt-101 as the teacher, and perform distillation based on VID. Its result is compared with other detector-specific distillation methods as below.
> > > | Student             | Teacher(s)                                  | Method                                | AP   |
> > > |---------------------|---------------------------------------------|---------------------------------------|------|
> > > | RetinaNet/ResNet-50 | RetinaNet/ResNeXt-101                       | VID [6]                              | 40.3 |
> > > | RetinaNet/ResNet-50 | RetinaNet/ResNeXt-101                       | CWD                                   | 40.8 |
> > > | RetinaNet/ResNet-50 | RetinaNet/ResNeXt-101                       | FGD                                   | 40.7 |
> > > | RetinaNet/ResNet-50 | RetinaNet/ResNeXt-101                       | MGD                                   | 41.0 |
> > > | RetinaNet/ResNet-50 | RetinaNet/ResNet-101->RetinaNet/ResNeXt-101 | MGD + Progressive distillation (Ours) | 41.4 |
> > >
> > > In this distillation setting with RetinaNet object detectors, VID fails to achieve comparable performance with detector-specific methods such as CWD, FGD, and MGD, and falls even farther behind MGD combined with our progressive distillation. As we have discussed previously, we believe that object detection introduces additional complexity to this distillation task, so it requires some customized mechanisms for successful knowledge transfer from the teacher detector to the student detector. As a result, a general-purpose distillation loss like VID may not be the optimal solution to our detector distillation task.
> > >
> > > Implementation details: This experiment is based on MMDetection, MGD, and a publicly available implementation of VID (https://github.com/HobbitLong/RepDistiller/blob/master/distiller_zoo/VID.py). The VID loss is computed at every level of the teacher’s and student’s FPN. The total loss to be minimized is the sum of the standard detection loss based on the ground-truth labels and $2.0\times$ the VID loss, where the balancing weight $2.0$ is selected from $\{0.5, 1.0, 2.0, 4.0, 6.0\}$ by maximizing the validation performance.
> > >
> > > We hope this experiment can address the reviewer’s concern. Please let us know if the reviewer has further questions, and we are very happy to discuss.

---

> > > ### Author Response · Authors · 2022-12-06
> > > **Update: Transformer-based Student**
> > >
> > > We thank the reviewer for suggesting distillation experiments on a Transformer-based student. As we have explained in the previous response, a convolution-based student is preferred for knowledge distillation in practice due to its efficiency. Therefore, in the submission we focused on experimenting with transferring knowledge from high-capacity Transformer-based teachers into a convolution-based student.
> > >
> > > Here, to answer the reviewer’s question and further test the generalizability of our proposed strategy of progressive distillation, we investigated distillation from convolution-based teachers into a Transformer-based student. The student is a RetinaNet/Swin-Tiny detector. As for the teachers, we choose detectors with the very recent, strong, convolution-based backbone: ConvNeXt [Ref1].  We use the state-of-the-art distillation loss MGD as in the submission. The comparison of the results are shown below:
> > >
> > > | Student             | Method                                | Teacher(s)                                                  | Schedule | AP   |
> > > |---------------------|---------------------------------------|-------------------------------------------------------------|----------|------|
> > > | RetinaNet/Swin-Tiny | No distillation                       | None                                                        | 2x       | 43.7 |
> > > | RetinaNet/Swin-Tiny | MGD + Direct distillation             | Cascade Mask R-CNN/ConvNeXt-Small                           | 2x       | 44.6 |
> > > | RetinaNet/Swin-Tiny | MGD + Progressive distillation (Ours) | Mask R-CNN/ConvNeXt-Tiny->Cascade Mask R-CNN/ConvNeXt-Small | 1x+1x    | 45.2 |
> > >
> > > As shown in this table, MGD direct distillation using a strong ConvNeXt-Small backbone gives the Transformer-based RetinaNet/Swin-Tiny student a 0.9% AP improvement. Our proposed progressive distillation, via an intermediate ConvNeXt-Tiny teacher, further boosts the student performance by 0.6% AP and leads to a high-accuracy student with 45.2% AP on COCO detection. This experiment again demonstrates the effectiveness of our progressive distillation across various teacher/student architectures, in addition to the Transformer-teacher, convolution-student setting which we showed in the submission. We will include this result in the revision.
> > >
> > > We hope this experiment, in addition to our previous response of the additional experiment on prior general KD methods (posted on November 24), can address the remaining concerns of the reviewer. Please let us know if there are any other questions. We would be very happy to address them.
> > >
> > > [Ref1] Zhuang Liu, Hanzi Mao, Chao-Yuan Wu, Christoph Feichtenhofer, Trevor Darrell, Saining Xie. A ConvNet for the 2020s. In CVPR, 2022.

---

### Official Review · Reviewer_XUDr · 2022-11-02

**Confidence:** 5
**Correctness:** 3
**Technical Novelty And Significance:** 3
**Empirical Novelty And Significance:** 3
**Recommendation:** 8

**Clarity, Quality, Novelty And Reproducibility:**

Good Clarity, Good Quality, Good Novelty, Poor Reproducibility. No codes are provided.

**Strength And Weaknesses:**

Strength:
1. This paper proposes to choose the policy of student-teacher in knowledge distillation by using the cost of linear regression on the validation set, which is novel in this domain,
2. Good experimental results have been achieved.
3. Sufficient ablation studies have been conducted to demonstrate their performance.

Weakness.
1. Most of the experiments are conducted on RetinaNet and Mask RCNN with different backbones. It will be better if results on more SOTA detection models can be provided, such as Deformable Detr, Yolov4, CenterNet and so on.
2. This paper applies the naive feature-based knowledge distillation as their knowledge distillation method. It is ok since this paper aims to focus on the order of teachers instead of the specific KD method. But It will be better if better more SOTA KD methods can be utilized to evaluate the effectiveness of the proposed method.

**Summary Of The Paper:**

This paper proposes a novel knowledge distillation methods which decide the progressive distillation process by the cost of linear regression on validation set, which achieves good performance on multiple student-teacher settings.

**Summary Of The Review:**

Please refer to the strength and weaknesses. In summary, I like the core idea of this paper, while some more experiments are still necessary.

Update-2022-11-14
The response from the authors basically addressed my concern. So I increase my score to 8.

---

> ### Author Response · Authors · 2022-11-08
> **Response to Reviewer XUDr**
>
> We appreciate the reviewer for the insightful comments and suggestions. Here we address all the concerns.
>
> - Detection models other than RetinaNet and Mask R-CNN:
>
>     In principle, our work is general and applicable to different types of detection models, as our progressive distillation strategy is designed without assumption on particular types of detector architectures.
>     - We chose RetinaNet and Mask R-CNN for a fair comparison with existing methods, because most of the prior work on detector distillation has primarily focused on RetinaNet and Mask R-CNN. Being commonly-used in real-world applications, RetinaNet and Mask R-CNN represent the two important families of object detectors (single-stage and two-stage) – most of the additional detectors suggested by the reviewer belong to either of these two families.
>     - As mentioned by **Reviewers 1tUC and zfUg**, in the original submission we also included results with state-of-the-art transformer-based detectors, in addition to convolution-based detectors. Please refer to Section 4.4 and Table 5, where we distill from teachers with Swin Transformer backbones.
>     - We agree with the reviewer that the evaluation would be more comprehensive, if we include more recently developed detectors. We are currently experimenting with some of these detectors, and will include the results in the final revision. If time allows, we will provide them by the end of this discussion period.
>
> - SOTA KD methods:
>
>     We agree with the reviewer, and indeed, in the original submission we included the results of combining our progressive distillation strategy with **three SOTA KD methods** for object detection and achieved the new best results, as shown in Section 4.4, Figure 4, and Table 5. The three leading methods are also feature-based distillation, but they utilize advanced mechanisms (e.g., feature generation, global-local attention) to fully exploit teachers’ features.
>
>     In addition, we would like to further clarify that our experiment with naive feature-based distillation is to demonstrate: after augmented by our progressive distillation strategy, the performance of the very naive feature-based distillation method can be dramatically improved, even outperforming some recent sophisticated distillation methods. For example, in Table 3 we showed a distilled RetinaNet/ResNet-50 student with 39.9 AP on COCO, outperforming [Ref1] (39.1 AP, CVPR 2021), [Ref2] (39.7 AP, CVPR 2021), and [Ref3] (39.6 AP, ICLR 2021). These results underscore the effectiveness of our progressive distillation strategy.
>
> - Code:
>
>     The code for reproducing our work was provided in the original supplementary material.
>
> Thank you again for your valuable review! We are looking forward to your further feedback.
>
> [Ref1] Xing Dai, Zeren Jiang, Zhao Wu, Yiping Bao, Zhicheng Wang, Si Liu, and Erjin Zhou. General instance distillation for object detection. In CVPR, 2021.
>
> [Ref2] Jianyuan Guo, Kai Han, Yunhe Wang, Han Wu, Xinghao Chen, Chunjing Xu, and Chang Xu. Distilling object detectors via decoupled features. In CVPR, 2021.
>
> [Ref3] Linfeng Zhang and Kaisheng Ma. Improve object detection with feature-based knowledge distillation: Towards accurate and efficient detectors. In ICLR, 2021.

---

> > ### Author Response · Authors · 2022-11-18
> > **Update: Additional Experiments with Detectors Other than RetinaNet or Mask R-CNN**
> >
> > We appreciate the reviewer for the helpful discussion and suggestions, and we are happy to see our previous response has addressed the reviewer’s concerns. Here we provide an update regarding the experiment previously requested by the reviewer. In our initial submission, we conducted experiments for RetinaNet and Mask R-CNN student detectors. The reviewer suggested results on more recent detection models.
> >
> > We have performed an additional experiment on another detection model, RepPoints [Ref4]. RepPoints is an anchor-free detector, with high efficiency-accuracy trade-off. For example, when equipped with the same ResNet-50 backbone, RepPoints performs better than CenterNet suggested by the reviewer (38.6 AP vs. 34.9 AP). Meanwhile, RepPoints has been experimented in state-of-the-art detector distillation methods such as MGD and FGD, so we can provide an informative comparison. Therefore, we choose RepPoints as the detection model in this additional experiment. The student model is RepPoints/ResNet-50 (38.6 AP on COCO), and following state-of-the-art method MGD, we use RepPoints/ResNeXt-101 (44.2 AP) as the final teacher model. The intermediate teacher in our progressive distillation is RepPoints/ResNet-101 (40.5 AP). The following table shows the results obtained.
> > | Model               | Method                                | AP       | AP$_S$ | AP$_M$ | AP$_L$ |
> > |---------------------|---------------------------------------|----------|--------|--------|--------|
> > | RepPoints/ResNet-50 | No distillation                       | 38.6     | 22.5   | 42.2   | 50.4   |
> > | RepPoints/ResNet-50 | MGD + Direct distillation             | 42.3     | 24.4   | 46.2   | 55.9   |
> > | RepPoints/ResNet-50 | MGD + Progressive distillation (Ours) | **42.9** | 25.6   | 46.9   | 56.3   |
> >
> > Our progressive distillation strategy improves the performance of the student detector by **4.3%** AP, which is also **0.6%** AP better than the previous state-of-the-art MGD. This result is achieved by only introducing an intermediate teacher and without increasing the training cost. This experiment further demonstrates that our progressive distillation strategy is general and can be applied to various detectors.
> >
> > [Ref4] Ze Yang, Shaohui Liu, Han Hu, Liwei Wang, and Stephen Lin. Reppoints: Point set representation for object detection. In ICCV, 2019.

---

### Official Review · Reviewer_df3Q · 2022-11-16

**Confidence:** 5
**Correctness:** 2
**Technical Novelty And Significance:** 2
**Empirical Novelty And Significance:** 2
**Recommendation:** 5

**Clarity, Quality, Novelty And Reproducibility:**

This paper addresses somehow an interesting problem of teacher selection. However, it does don't make enough sense of why selecting multiple transformer models and distill them to learn a CNN student. The selection strategy of teacher models lacks flexibility as it should be based on a "proper" selection and measure for successful distillation. From this, we can see that the method itself is not very intuitive and persuasive. What if bad teachers can still be used for learning a good student? References, such as [1,2,3], can be good examples.

[1] Knowledge Distillation: Bad Models Can Be Good Role Models"
[2] Student Customized Knowledge Distillation: Bridging the Gap Between Student and Teacher
[3] Distilling the Undistillable: Learning from a Nasty Teacher

**Strength And Weaknesses:**

(1) This paper is well-written and organized.

(2) The idea of learning lightweight object detectors via progressive knowledge distillation and investigating heterogeneous knowledge distillation is interesting.

(3) The experimental results significantly outperform the existing SOTA knowledge distillation methods.


Weakness:

(1) The authors failed to prove that the observation ‘teacher-student capacity gap can be solved by the sequential distillation from multiple teachers arranged into a curriculum’, which is claimed in the Introduction.

(2) Why the heterogeneous knowledge distillation should be performed from the Transformer-based teachers to convolution-based students? Why not from the convolution-based teachers to Transformer-based students?

(3) Is the multiple teachers necessary? How about utilizing a single-teacher network with multiple forward progress? This operation also gives more supervision to the selected student and meanwhile saves the effort of searching for the near-optimal teacher order. Authors need to show the rationality of multiple teachers rather than multiple forward passes.

(4) More SOTA and widely-adopted detection models should be involved, eg., YoloV4.

(5) This paper repeatedly mentioned about "proper" either in selecting the teacher network or 'arranging teacher sequence'. Indeed, this reflected that this paper proposed a method that is not adaptable enough but based on "proper" (manual) selection and usage to
achieve good performance. It is actually not easy to be extended by the followers if published.

(6) As mentioned, selecting multiple teachers is actually not very easy in reality, especially for Transformers, which can not be easily trained. But it is more approachable to transfer from CNNs to the transformer as CNNs are already matured.  Therefore, the rationality of this method is less clear.

(7) I do agree that teacher selection is one important aspect, however, currently, the method seems to lack enough novelty and complexity to meet the acceptance bar of ICLR.


**Summary Of The Paper:**

This paper provides a simple yet effective sequential approach to distill knowledge from Transformer-based teacher detectors to convolution-based student detectors. Instead of learning from a single teacher, the proposed framework develops a principled way to automatically design a sequence of teachers appropriate for the student and progressively distill it. Notably, the authors claimed that they were the first to investigate distillation from Transformer-based teacher detectors to a convolution-based student. Extensive experiments and ablation study is given to show the effectiveness of the proposed method.

**Summary Of The Review:**

This paper is somewhat interesting but lacks rationality of clear motivation, technical novelty, and critical ablation studies. Details can be referred to the strength and weakness

---

> ### Author Response · Authors · 2022-11-20
> **Response to Reviewer df3Q (part 1)**
>
> We appreciate the reviewer for the insightful comments and suggestions. Here we address all the concerns.
>
> - First of all, we would like to clarify that our proposed progressive distillation is a **general-purpose** strategy for object detectors, not just for distillation “from Transformer-based teacher detectors to convolution-based student detectors.” In the experiment section, we showed a wide range of distillation settings including multiple convolution-based teachers and students (please see Table 1 where we included various convolution-based models in our experiments). The distillation experiment from Transformer-based teacher to convolution-based student is just one example to show the generalizability of our progressive distillation, in particular to test the extreme case where the backbones of the teacher and student are drastically different and exhibit a huge capacity gap due to this architectural difference.
>
> - The authors failed to prove that the observation ‘teacher-student capacity gap can be solved by the sequential distillation from multiple teachers arranged into a curriculum’, which is claimed in the Introduction.
>
>     We believe that our extensive experiments have empirically validated this. As mentioned by **Reviewer 1tUC**, “progressively transferring knowledge from a sequence of teachers to a lightweight detector is somewhat novel.” Also, we provide “promising results and convincing ablation studies with multiple settings for object detection” to support our claim as pointed out by **Reviewer R4rk**.
>
>     We would greatly appreciate it if the reviewer could further clarify why the reviewer thinks our progressive distillation using multiple teachers is invalid for bridging the “teacher-student capacity gap.” In Sections 4.2, 4.3, and 4.4, we consistently showed that our progressive distillation strategy is the key to improving the distilled student detector’s performance, when there is a large capacity gap between the teacher and the student. For example, in Figure 4-left, the performance of FGD-distilled RetinaNet/ResNet-50 student improves from 40.7% to 41.5% AP (+0.8%). This gain comes from our progressive distillation that uses an intermediate teacher (RetinaNet/ResNet-101) and then the final teacher (RetinaNet/ResNeXt-101). The intermediate teacher serves as the bridge between the high-capacity teacher and the low-capacity student to help the student progressively adapt.
>
>     We are happy to provide more explanation if the reviewer could provide some reasons that show our work fails to support the claim about the proposed progressive distillation strategy.

---

> > ### Author Response · Authors · 2022-11-20
> > **Response to Reviewer df3Q (part 2)**
> >
> > - Why not from the convolution-based teachers to Transformer-based students?
> >
> >     We did not distill Transformer-based students from convolution-based teachers in the original submission for the following reasons:
> >
> >     - Consistent with practical scenarios: The end goal of this work and knowledge distillation in general is to develop high-accuracy, lightweight object detectors. To this end, the student detectors are typically efficient detectors like RetinaNet with a ResNet backbone. Although there has been some recent research focusing on designing computationally efficient architectures for Transformers, the widely adopted vision Transformers (e.g., ViT and Swin Transformer) are still slower than their convolution-based counterparts (e.g., ResNet and EfficientNet), because convolution operations are highly optimized on GPUs. Therefore, ResNet-based students are more favored in developing lightweight object detectors and are thus used in our evaluation.
> >
> >         Meanwhile, the reviewer pointed out that “it is more approachable to transfer from CNNs to the transformer as CNNs are already matured.” However, when developing lightweight object detectors, we usually care less about the training efforts of the teachers, as long as the students are computation-efficient and accurate at test time. The setting where distillation happens from high-capacity Transformer teachers to low-capacity convolution-based students is thus more challenging and more important for real-world applications, than the case of distillation from convolution-based teachers to Transformer-based students.
> >
> >     - The key question we want to address is the **capacity gap** between the student and the teacher. Typically, commonly used Transformer-based backbones (e.g., Swin-Small) have stronger performance than convolution-based backbones (e.g., ResNet-101), and thus possess a higher learning capacity. We study the **more challenging** case of distilling knowledge from a high-capacity Transformer-based teacher (for better performance) to a low-capacity convolution-based student (for better efficiency), so the capacity gap is unavoidable and must be mitigated. In such a scenario, our progressive distillation is a helpful solution. In fact, we showed progressive distillation is critical to the success of knowledge transfer from a Swin Transformer teacher to a ResNet student in Table 5.
> >
> >
> > - Why not one single teacher network with “multiple forward passes”?
> >
> >     We would greatly appreciate it if the reviewer could further clarify about “multiple forward passes.” If that refers to distillation from only one teacher to the student with more training iterations, that is exactly the baseline of direct distillation in Table 3 (ID 3 & 7), Table 5 (ID 1 & 3), and Figure 4 (blue bars). Please note that we keep the total training iterations the same between direct distillation and our progressive distillation for fair comparison, following the standard practice in previous detector distillation methods like MGD and FGD. If only one single teacher (let’s call it model A) is used in distillation, we use a “2x” (24 epochs on COCO) training schedule. If there are two teachers (let’s call them models B->A) that progressively distill the student, we use a “1x” (12 epochs on COCO) training schedule for each teacher. That means, in the former case, the single teacher A is utilized with **doubled training iterations** (“multiple forward passes”) than teacher A in the latter case.
> >
> >     If we understand “multiple forward passes” incorrectly from the reviewer, we sincerely request the reviewer to provide more clarification for further discussion and experiments.

---

> > > ### Author Response · Authors · 2022-11-20
> > > **Response to Reviewer df3Q (part 3)**
> > >
> > > - More detection models:
> > >
> > >     Per the reviewer’s request, we have performed an additional experiment on another detection model, RepPoints [Ref1]. RepPoints is an anchor-free detector, with high efficiency-accuracy trade-off. Meanwhile, RepPoints has been experimented in state-of-the-art detector distillation methods such as MGD and FGD, so we can provide an informative comparison. Therefore, we choose RepPoints as the detection model in this additional experiment. The student model is RepPoints/ResNet-50 (38.6 AP on COCO), and following state-of-the-art method MGD, we use RepPoints/ResNeXt-101 (44.2 AP) as the final teacher model. The intermediate teacher in our progressive distillation is RepPoints/ResNet-101 (40.5 AP). The following table shows the results obtained.
> > >
> > >     | Model               | Method                                | AP       | AP$_S$ | AP$_M$ | AP$_L$ |
> > >     |---------------------|---------------------------------------|----------|--------|--------|--------|
> > >     | RepPoints/ResNet-50 | No distillation                       | 38.6     | 22.5   | 42.2   | 50.4   |
> > >     | RepPoints/ResNet-50 | MGD + Direct distillation             | 42.3     | 24.4   | 46.2   | 55.9   |
> > >     | RepPoints/ResNet-50 | MGD + Progressive distillation (Ours) | **42.9** | 25.6   | 46.9   | 56.3   |
> > >
> > >     Our progressive distillation strategy improves the performance of the student detector by **4.3%** AP, which is also **0.6%** AP better than the previous state-of-the-art MGD. This result is achieved by only introducing an intermediate teacher and without increasing the training cost. This experiment further demonstrates that our progressive distillation strategy is general and can be applied to various detectors.
> > >
> > >
> > > - “Proper” selection of multiple teachers, technical novelty:
> > >
> > >     We would like to clarify that we did develop a principled algorithm for **automatically** selecting and arranging teachers from multiple candidates, as described in Section 3.2 and Algorithm 1 in Appendix. Section 4.1 validates that our Algorithm 1 provides near-optimal teacher orders, by comparing them with all possible teacher orders (note that extensively enumerating all orders is not necessary in applications). This algorithm is based on the representation similarities between detection models. With this automatic approach for selecting teachers and designing a teacher sequence, one can readily extend progressive distillation to other pools of teacher detector candidates, without relying on manual selection. For example, we used the automatically generated teacher orders from different teacher pools for two students (RetinaNet and Mask R-CNN) in Section 4.2.
> > >
> > >     The novelty of our approach has also been acknowledged by other reviewers. As mentioned by **Reviewer XUDr**, our paper “proposes to choose the policy of student-teacher in knowledge distillation by using the cost of linear regression on the validation set, which is novel in this domain.” Also, as mentioned by **Reviewer zfUg**, “the sequential strategy is inspiring and of interest for knowledge distillation for object detection and segmentation.”
> > >
> > >     We are happy to provide more explanation if the reviewer has further concerns regarding this algorithm for automatically selecting teacher orders.
> > >
> > > [Ref1] Ze Yang, Shaohui Liu, Han Hu, Liwei Wang, and Stephen Lin. Reppoints: Point set representation for object detection. In ICCV, 2019.

---

> > > > ### Author Response · Authors · 2022-11-20
> > > > **Response to Reviewer df3Q (part 4)**
> > > >
> > > > - Other related work [1][2][3]:
> > > >
> > > >     We appreciate the reviewer for pointing out these related previous methods on knowledge distillation from teachers which may not be directly beneficial due to noise or capacity mismatch:
> > > >
> > > >     - [1][3] are not directly applicable in our setting for two reasons:
> > > >
> > > >         - They focus on the image classification task rather than object detection. The detection task (and the associated model architectures) is much more complicated than the classification task. This makes the distillation methods developed in the context of classification often not directly applicable to detection. That is why dedicated distillation methods need to be developed for the detection task in the literature. We would greatly appreciate the reviewer to take this important distinction when comparing our work with prior work.
> > > >
> > > >         - They are based on “bad” teachers, which are trained from noisy data [1] or an adversarial loss [3]. However, in our setting of developing lightweight object detectors, we do not have the concerns about noisy data [1] or private data [3], so it is more natural to simply use normally trained teachers with higher performance for distillation.
> > > >
> > > >     - [2] examines the capacity mismatch between teacher and student from the perspective of gradient similarity. The feature distillation (per level) is activated only when the gradient is similar between the teacher and student. This work also focuses on one single teacher, which is fundamentally different from ours. We believe this adaptive strategy is orthogonal to our work that uses multiple teachers to address the capacity mismatch issue. Studying the interaction between these two strategies is an excellent direction for future research.
> > > >
> > > > Thank you again for your valuable review! We are looking forward to your further feedback.

---

> > > ### Author Response · Authors · 2022-12-06
> > > **Update: Transformer-based Student**
> > >
> > > We thank the reviewer for suggesting distillation experiments on a Transformer-based student. As we have explained in the previous response, a convolution-based student is preferred for knowledge distillation in practice due to its efficiency. Therefore, in the submission we focused on experimenting with transferring knowledge from high-capacity Transformer-based teachers into a convolution-based student.
> > >
> > > Here, to answer the reviewer’s question and further test the generalizability of our proposed strategy of progressive distillation, we investigated distillation from convolution-based teachers into a Transformer-based student. The student is a RetinaNet/Swin-Tiny detector. As for the teachers, we choose detectors with the very recent, strong, convolution-based backbone: ConvNeXt [Ref1].  We use the state-of-the-art distillation loss MGD as in the submission. The comparison of the results are shown below:
> > >
> > > | Student             | Method                                | Teacher(s)                                                  | Schedule | AP   |
> > > |---------------------|---------------------------------------|-------------------------------------------------------------|----------|------|
> > > | RetinaNet/Swin-Tiny | No distillation                       | None                                                        | 2x       | 43.7 |
> > > | RetinaNet/Swin-Tiny | MGD + Direct distillation             | Cascade Mask R-CNN/ConvNeXt-Small                           | 2x       | 44.6 |
> > > | RetinaNet/Swin-Tiny | MGD + Progressive distillation (Ours) | Mask R-CNN/ConvNeXt-Tiny->Cascade Mask R-CNN/ConvNeXt-Small | 1x+1x    | 45.2 |
> > >
> > > As shown in this table, MGD direct distillation using a strong ConvNeXt-Small backbone gives the Transformer-based RetinaNet/Swin-Tiny student a 0.9% AP improvement. Our proposed progressive distillation, via an intermediate ConvNeXt-Tiny teacher, further boosts the student performance by 0.6% AP and leads to a high-accuracy student with 45.2% AP on COCO detection. This experiment again demonstrates the effectiveness of our progressive distillation across various teacher/student architectures, in addition to the Transformer-teacher, convolution-student setting which we showed in the submission. We will include this result in the revision.
> > >
> > > We hope this experiment, in addition to our previous response (posted on November 20) to the official review, can address the concerns raised by the reviewer. Please let us know if there are any other remaining questions. We would be very happy to address them.
> > >
> > > [Ref1] Zhuang Liu, Hanzi Mao, Chao-Yuan Wu, Christoph Feichtenhofer, Trevor Darrell, Saining Xie. A ConvNet for the 2020s. In CVPR, 2022.

---

### Author Response · Authors · 2022-11-20
**Paper Revision**

Dear All,

We greatly appreciate all the reviewers for their valuable suggestions and comments. In addition, we thank Reviewers XUDr and zfUg for acknowledging our response and re-evaluating our work.

Based on the feedback from all the reviewers, we have uploaded a revised version of our work. The changed parts are marked in blue. Due to space limitations, the revised parts are added to our Appendix. In this revision, we have mainly included:

- More detailed discussion on prior knowledge distillation methods

- Two experimental results requested by reviewers: another detector model, and ablation on keeping the intermediate teacher

- More implementation details

We hope this revised version can help everyone address their concerns about our work.

Best,

Authors

---

### Decision · Program_Chairs · 2023-01-20

**Decision:**

Reject

**Justification For Why Not Higher Score:**

Reviewers have reached the consensus that the technical novelty of this paper is limited, given the vast amount of closely relevant work listed by multiple reviewers.

**Justification For Why Not Lower Score:**

N/A

**Metareview: Summary, Strengths And Weaknesses:**

The submission introduces a sequential approach to knowledge distillation that progressively transfers the knowledge of a set of teachers to a lightweight student. Reviewers overall like the good results but are concerned about the technical novelty of the work. Post rebuttal, there is a lot of discussion about this submission.  AC and reviewers discussed the paper on Open Review (which are not visible to authors or public), and the AC invites both positive and negative reviewer to comment and exchange opinions. Reviewers have reached the consensus that the technical novelty of this paper is limited, given the vast amount of closely relevant work listed by multiple reviewers.  Reviewer XUDr was the most positive about the submission; unfortunately, the AC was not able to get the reviewer to participate in any discussion or respond to the novelty concern.  However, XUDr's updated comments and the authors' note to AC were all taken into consideration.

Considering all the pros and cons, the AC recommends rejection so that the authors will be able to systematically address the concerns on novelty. The authors are encouraged to clearly state and demonstrate how this submission adds to the extensive literature on knowledge distillation.